# Touch-sensitive stamens enhance pollen dispersal by scaring away visitors

**Deng-Fei Li[1], Wen-Long Han[1], Susanne S Renner[2], Shuang-Quan Huang[1]\***

[1]Institute of Evolution and Ecology, School of Life Sciences, Central China Normal University, Wuhan, China; [2]Department of Biology, Washington University, Saint Louis, United States

**Abstract** Animal-pollinated plants have to get pollen to a conspecific stigma while protecting it from getting eaten. Touch-sensitive stamens, which are found in hundreds of flowering plants, are thought to function in enhancing pollen export and reducing its loss, but experimental tests are scarce. Stamens of *Berberis* and *Mahonia* are inserted between paired nectar glands and when touched by an insect's tongue rapidly snap forward so that their valvate anthers press pollen on the insect's tongue or face. We immobilized the stamens in otherwise unmodified flowers and studied pollen transfer in the field and under enclosed conditions. On flowers with immobilized stamens, the most common bee visitor stayed up to 3.6× longer, yet removed 1.3× fewer pollen grains and deposited 2.1× fewer grains on stigmas per visit. Self-pollen from a single stamen hitting the stigma amounted to 6% of the grains received from single bee visits. Bees discarded pollen passively placed on their bodies, likely because of its berberine content; nectar has no berberine. Syrphid flies fed on both nectar and pollen, taking more when stamens were immobilized. Pollen-tracking experiments in two *Berberis* species showed that mobile-stamen-flowers donate pollen to many more recipients. These results demonstrate another mechanism by which plants simultaneously meter out their pollen and reduce pollen theft.

**\*For correspondence:**
hsq@ccnu.edu.cn

**Competing interest:** The authors declare that no competing interests exist.

## Editor's evaluation

With a series of manipulative experiments using four plant species with stamens that can snap toward the stigma if touched at the base, the authors provide compelling evidence that pollinators stay longer yet export less pollen to recipient flowers when stamens are immobilized by alcohol application. This is a landmark study on the functional consequences and adaptive significance of a phenomenon scattered throughout the angiosperm clade.

## Introduction

Animals visit flowers to forage for food or other rewards, mainly nectar or pollen (*Ollerton, 2021*). From the perspective of male reproductive success, nectar and pollen are entirely different rewards because paternity is maximized if pollen grains from one flower are deposited on multiple conspecific recipients, rather than ending up as food, while nectar is produced as pollinator food (*Westerkamp, 1996*; *Westerkamp and Classen-Bockhoff, 2007*). A plant's success as a father can depend on its temporal deployment of pollen and on the accuracy of pollen placement on the most effective pollen vectors (*Harder and Thomson, 1989*; *Armbruster et al., 2014*, and studies cited therein). Flowers are therefore under selection to 'pay' visitors as much as possible by nectar, which can usually be replenished, and to meter out their non-replenishable pollen grains by placing them on multiple high-fidelity vectors. Two ways in which plants achieve this are by filtering their visitors, such that pollen

**Figure 1.** Flower traits, foraging behavior of visitors, and manipulations of stamen movements in *Berberis julianae*, which has stamens characterized by a touch-sensitive rapid movement toward the flower center. The major pollinators, workers of *Apis cerana* (**A**), and a long-tongued bee, *Habropoda sichuanensis* (**B**), sucking nectar while their tongues (arrow) may contact filaments, anthers, and/or stigmas. These bees do not groom *Berberis* pollen into their corbiculae, and their legs are therefore without pollen loads (hollow arrows). (**C**) *Rhingia campestris* feeding on nectar and pollen. A bee visiting two flowers with experimentally immobilized and hence touch-insensitive stamens (**D**). (**E**) A cross section of a floral bud, showing the two anther valves and two nectaries at the base of each petal. (**F**) Natural flower with mobile stamens (left) bending inward when their filament bases are touched by a needle; stamen-immobilized (SI) flower (right) whose pedicel had been immersed in 75% alcohol for over 30 min. (**G**) Diagram of stamen-mobile and SI flowers, illustrating the stamen movement when a bee's tongue touches the filament. (**H**) A floral array on an inflorescence in the field with four alcohol-treated SI flowers (arrows). (**I**) Stained pollen grains (red) deposited on a stigma under open pollination in the field.

The online version of this article includes the following figure supplement(s) for figure 1:

**Figure supplement 1.** Flower manipulations and experimental floral arrays in *Berberis julianae*.

**Figure supplement 2.** Habit, floral traits, developing berries, and feeding behavior of flower visitors in three Berberidaceae species whose stamens are touch-sensitive.

consumers are repelled, or by encouraging pollinators to move on, resulting in less pollen and/or nectar consumption per flower while retaining pollen import and export (male and female fitness).

Biologists from Linnaeus (1755) onward have been aware of the forward-snapping movement of the stamens of *Berberis vulgaris* once the filament base of an individual stamen is touched by a nectar-drinking insect (or a pointed object). *Berberis* flowers have six petals and six stamens, each inserted between two nectaries. Nectar constitutes the main floral reward. The relatively few pollen grains produced remain hidden in the paired pollen sacs that open by apical valves (*Figure 1*, *Supplementary file 1*; pollen/ovule ratios in *Berberis* are around 2380–3400; this study). Different species of

*Berberis* and *Mahonia*, a close relative, vary in the rapidity of their stamen movements and also in the extent of recovery and repeatability, but the first movement of single stamen is generally completed in fractions of seconds (*Percival, 1965*; *Lechowski and Białczyk, 1992*; *Lebuhn and Anderson, 1994*; this study).

Early workers thought that the unidirectional stamen movements to the flower center, where the stigma is located, played a role in self-pollination (e.g., *Linnaeus, 1755*; *Sprengel, 1793*), but since about the 1880s, it has generally been assumed that the stamen movement helps to precisely pack pollen on the tongues or faces of flies or bees (*Kirchner, 1911*; *Knoll, 1956*; *Percival, 1965*; *Kugler, 1970*; *Lebuhn and Anderson, 1994*). Kirchner (*Kirchner, 1911*) furthermore suggested that insects hit by a stamen would be encouraged to leave the flower, but soon would land on another flower to resume their nectaring. Rapid succession of brief visits to many flowers in Kirchner's view should enhance cross-pollination and reduce nectar costs per pollen grain transport. A third possibility would be the above-mentioned visitor filtering if different kinds of insects were to react differently to forward-snapping of the stamens. This is the case, for example, in species of *Opuntia*, where forward-moving stamens make the pollen almost inaccessible to generalist bees and near-exclusively accessible to specialized bees that pollinate the flowers (*Schlindwein and Wittmann, 1997*).

In this study, we experimentally test these three hypotheses (filtering visitor species; metering out small numbers of pollen grains onto body parts likely to come in contact with conspecific stigmas; making pollinators leave more quickly) by immobilizing the stamens in the flowers of three species of *Berberis* and one species of *Mahonia*. Filament bending in *Berberis* relies on rapid changes in the calcium permeability of membranes (*Lechowski and Białczyk, 1992*), and we therefore explored treatments with calcium inhibitors and with alcohol. We discovered that immersion of flower pedicles in 75% alcohol for 35–45 min was effective at blocking the stamen movement. A test for possible effects of the alcohol treatment on foraging behaviors of the major floral visitors revealed no statistical effects. We then built experimental arrays with untreated and treated flowers in enclosed conditions to quantify pollen export and import from single visits of bees and flies and also used pollen staining to track pollen export distances from manipulated and control flowers.

Our experiments allowed us to address the following questions: (1) Are *Berberis* flowers with mobile stamens visited by the same types of insects and at the same rates as flowers with immobilized stamens? Different visitor types or visitation rates are expected under the filtering hypothesis, for instance, with flies reacting differently to touch-sensitive stamens than bees. (2) Do forward-snapping stamens make pollinators leave more quickly, reducing nectar costs per pollen grain transport? (3) Do flowers with touch-sensitive stamens export more pollen to more flowers and/or flowers further away? Lastly, we tested whether berberine, an alkaloid with antifeedant activity against herbivores found in *Berberis* leaves (*Schmeller et al., 1997*; *Manosalva et al., 2019*), is also present in *Berberis* pollen or nectar.

## Results

### Touch-sensitive stamen movements and experimental selfing and outcrossing

Flowers of *Berberis* and *Mahonia* have six petals and six stamens inserted between paired nectar glands (organ sizes are given in *Supplementary file 1*). Each pollen sac opens by a separate valve and contains about 610 ± 6 sticky yellow pollen grains that remain attached to the pollen sac (*Figure 1*, *Figure 1—figure supplement 1A*, *Figure 1—figure supplement 2*). When a flower visitor (or a pointed object) contacts the adaxial surface of a filament base, the respective stamen rapidly moves toward the flower center, placing pollen grains on the visitor's tongue (*Figure 1A and B*, *Figure 1—figure supplement 2E and F*). The stamen movement takes 0.44 ± 0.02 s in *Berberis julianae*, 0.17 ± 0.02 s in *B. jamesiana*, and 0.23 ± 0.04 s in *B. forrestii* (*Supplementary file 1*). Within 1 min, the stamen moves back from the flower center, where the single style with its large stigma is located, to the petal, taking 227.70 ± 10.06 s, 110.37 ± 6.64 s, and 155.31 ± 14.07 s, respectively, to return its original position (*Supplementary file 1*). In *Mahonia bealei*, the stamen movement takes 0.09 ± 0.01 s (N = 10 flowers), and 3.46 ± 0.71 s later, the stamen starts moving back, taking 7.74 ± 1.96 s to return to its original position (Video 2).

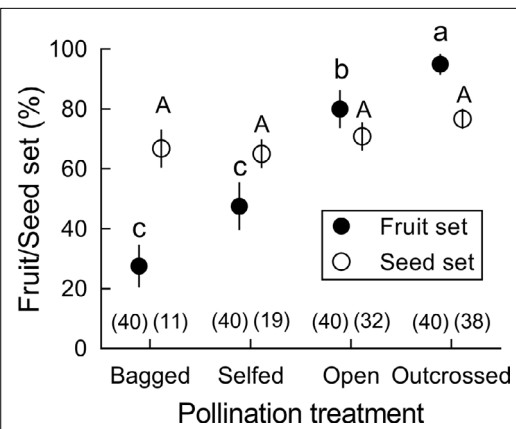

**Figure 2.** Fruit and seed set (mean ± Standard Error) after four pollination treatments in *Berberis julianae*. Different letters beside mean values indicate significant differences among the four treatments under a generalized linear model (GLM). Fruit set differed significantly (Wald $\chi^2$ = 34.598, p<0.001, df = 3) but seed set per fruit did not when zero data were excluded (Wald $\chi^2$ = 1.973, p=0.578, df = 3). Sample size for each treatment is given in brackets above the X-axis.

In *B. julianae*, fruit set in open-pollinated flowers (80.0 ± 6.4%) was significantly higher than in bagged and self-fertilized flowers (27.5 ± 7.1%; *Figure 2*). However, seed set per fruit did not differ between manually self-pollinated and cross-pollinated flowers. Self-pollen receipt by stigmas of *B. julianae* after a single stamen movement (14 ± 3 grains, N = 16) was only 6% of the pollen receipt resulting from a single visit by the most common visitors, *Apis cerana*, which deposited between 230 and 260 grains (section 'Effects of forward-snapping stamens on visitor behavior and pollination in the 22 field and under-enclosed conditions') and roughly 1% of the pollen grains of a single anther with its two pollen sacs (ca. 1220 grains), indicating that intra-flower self-pollination mediated by the stamen movements plays a minor role in total pollen receipt.

## Flower visitors and pollinators

At our study site, *B. julianae* was visited mainly by five insect species (*Supplementary file 2*), the bee *A. cerana* Fab., 1793 (*Figure 1A*), two anthophorid bees *Anthophora waltoni* Cockerell, 1910, and *Habropoda sichuanensis* Wu, 1986 (*Figure 1B*), and the syrphid flies *Rhingia campestris* Meigen, 1822, and *Meliscaeva* spec. In the field, it was not always possible to securely distinguish the anthophorid bees, although *H. sichuanensis* was clearly more frequent, and some of our results, for example, on visitation rates, therefore pool these species. These bees foraged for nectar, but not pollen, while the flies fed on both nectar and pollen (*Figure 1C*). Consistent with these feeding habits, pollen transfer efficiency of the bees was significantly higher than that of the flies (*Supplementary file 2*).

### Tests for a possible confounding effect of the alcohol treatment on visitor behavior

We found no effect of any lingering alcohol scents (in stamen-immobilized flowers) on visitor behavior: Visitation rates of *A. cerana* to *B. julianae* (under enclosed conditions) did not differ between untreated flowers with mobile stamens (SM), flowers with immobilized stamens (SI) via alcohol immersion, and untreated flowers in a fixed position above alcohol, called SMA flowers (Wald $\chi^2$ = 0.194, df = 2, p=0.908; *Figure 3A*). However, *A. cerana* stayed longer in SI compared to SM and SMA flowers (Wald $\chi^2$ = 64.599, df = 2, p<0.001; *Figure 3B*), showing that it was the forward-snapping stamen that caused these bees to leave. Visitation rates of *A. cerana* to *B. julianae* (again under enclosed conditions) also did not differ among SM, SI, and filament-damaged (FD) flowers (Wald $\chi^2$ = 0.44, df = 2, p=0.802; *Figure 3C*) because all three types of flowers offered the nectar sought by these bees. However, bees stayed less time in SM flowers than in flowers without stamen bending (SI and FD flowers) and equally long in SI and FD flowers (*Figure 3D*), showing that it was the stamen forward-snapping per se that caused visitors to move on.

### Effects of forward-snapping stamens on visitor behavior and pollination in the field and under enclosed conditions

In the field, when three or four SI flowers of *B. julianae* were inserted on racemes with the same number of SM flowers (*Figure 1H*), *A. cerana* stayed much longer on SI flowers than on SM flowers (15.46 ± 1.54 s vs. 3.63 ± 0.33 s, Wald $\chi^2$ = 56.055, p<0.001 in 2020; 16.076 ± 1.515 s vs. 5.675 ± 0.382 s, Wald $\chi^2$ = 68.421, p<0.001 in 2021). Despite the longer visits, fewer pollen grains were loaded onto *A. cerana* after a single visit to SI flowers than to SM flowers (716 ± 85 vs. 1223 ± 100

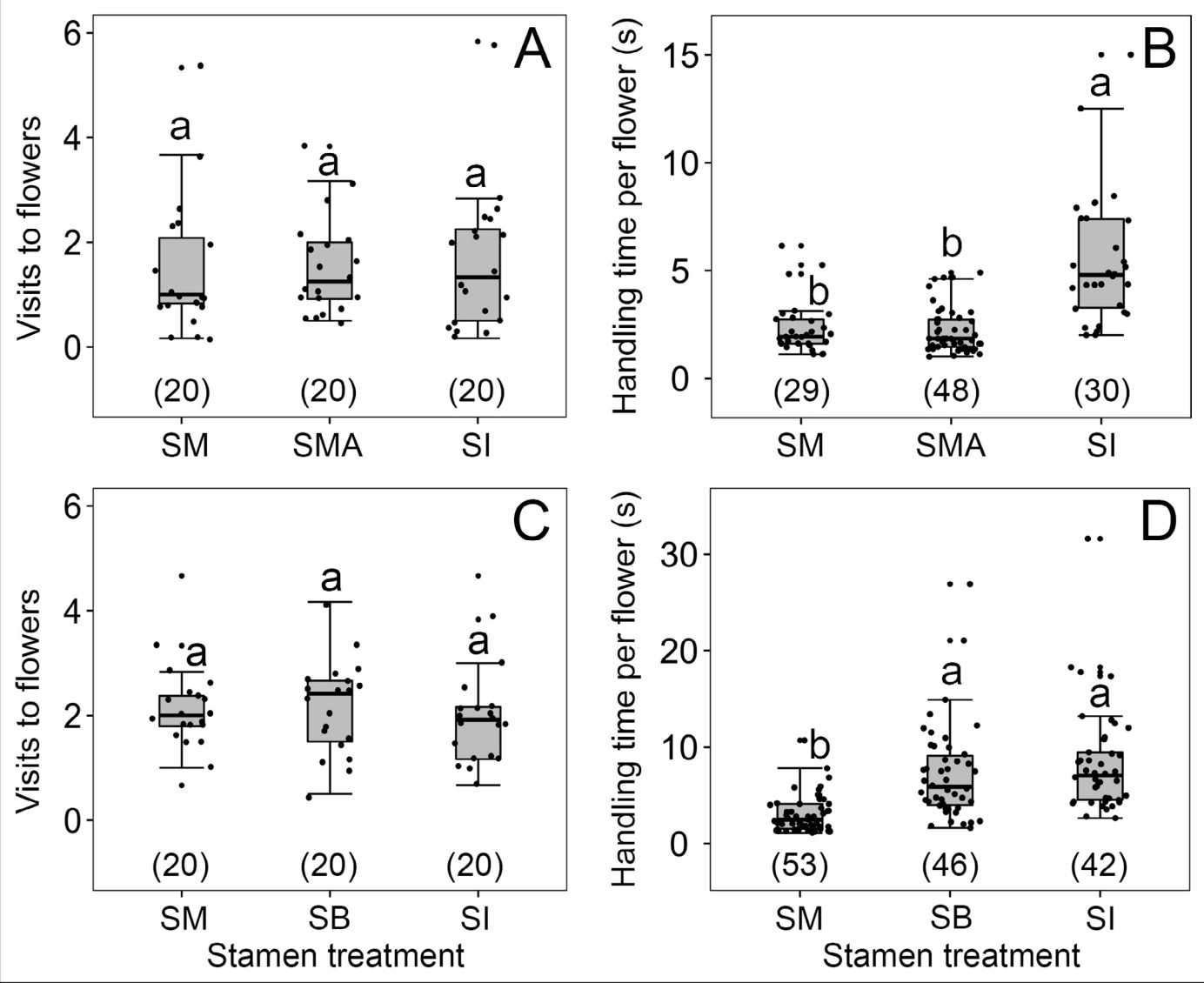

**Figure 3.** Visitation rates (**A, C**) and handling times (**B, D**) of *Apis cerana* in four treatments of flowers of *Berberis julianae*. Stamens mobile (SM, controls), stamens immobilized (SI), natural flowers in a fixed position above alcohol (SMA), and flowers with their filaments damaged (FD) so that the stamens became immobile but retained their pollen sacs and the nectar glands to the right and left of each filament. The box plots indicate the median (mid lines), the interquartile range (boxes), and 1.5× the interquartile range (whiskers). Different lowercase letters indicate significant differences, and the numbers in brackets indicate sample sizes. Sample sizes (n) are given in brackets.

grains in 2020; 890 ± 73 vs. 1401 ± 134 grains in 2021; Wald $\chi^2$ = 12.873, p<0.001 in 2020; Wald $\chi^2$ = 10.511, p=0.001 in 2021), and the numbers of pollen grains deposited on stigmas by a single *A. cerana* visit also were much lower in SI flowers than in SM flowers (87 ± 10 vs. 230 ± 25 grains in 2020; 63 ± 8 vs. 260 ± 35 grains in 2021; Wald $\chi^2$ = 26.847, p<0.001 in 2020; Wald $\chi^2$ = 40.042, p<0.001 in 2021; Figure 5, *Figure 5—figure supplement 1*). Pollen transfer efficiency by *A. cerana* was therefore reduced in SI compared to SM flowers, and this was significant in 2021 but not 2020 (0.084 ± 0.016 vs. 0.303 ± 0.102; Wald $\chi^2$ = 4.505, p=0.034 in 2021; 0.205 ± 0.024 vs. 0.246 ± 0.091; Wald $\chi^2$ = 0.188, p=0.665 in 2020; *Figure 5—figure supplement 1I and J*).

When the experiment was repeated under enclosed condition (using five SI and five SM flowers), the bees and syrphid flies all stayed longer on SI than on SM flowers (33.7 ± 4.2 s vs. 16.0 ± 1.7 s; Wald $\chi^2$ = 30.106, p<0.001; *Figure 4E'–T*). All visitor species exploited more nectar (*Figure 4M'–P*) and touched more stamens in SI than in SM flowers (*Figure 4I'–L*). The pollen transfer efficiency of the bees was higher than that of the flies (Wald $\chi^2$ = 13.319, df = 3, p=0.004, N = 80). In SI flowers, *A. cerana*

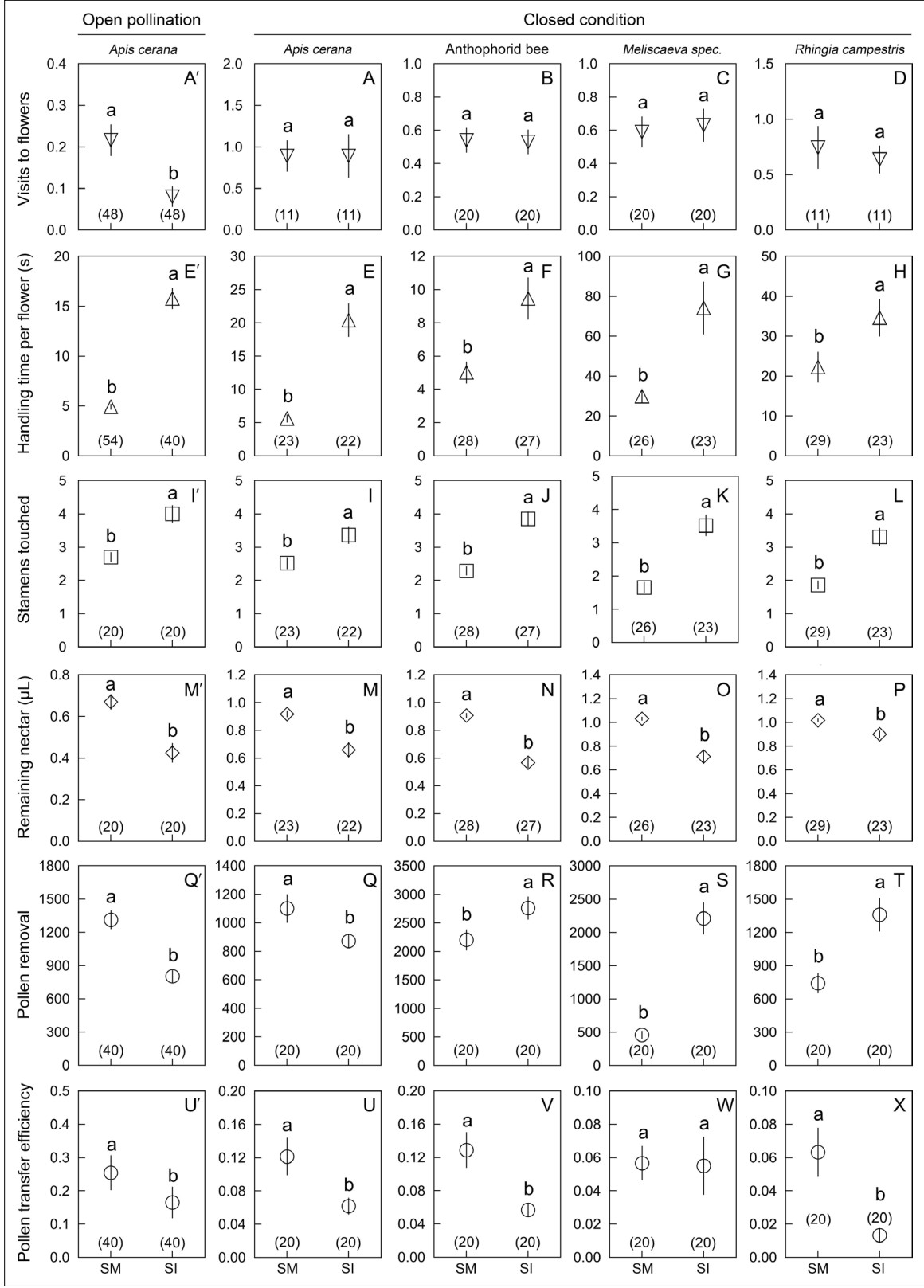

**Figure 4.** Comparisons of six parameters (mean ± Standard Error) in *Berberis julianae* to examine the effects of stamen movements on insect visitor foraging behavior and their roles in pollination. The major pollinator, *Apis cerana*, was studied in 2020 and 2021 under open pollination (far left) and enclosed conditions (set-up shown in *Figure 1—figure supplement 1* and Video 3), whereas the anthophorid bee pollinators and the two flies (*Meliscaeva* spec. and *Rhingia campestris*) were compared under enclosed conditions in 2021. Different lowercase letters above error bars indicate

*Figure 4 continued on next page*

*Figure 4 continued*

significant differences between control (stamen mobile [SM]) and alcohol-treated (stamen immobilized [SI]) flowers. (**A′–D**) Visitation rates of four visitor species, showing that *A. cerana* visited control flowers more frequently than SI flowers under open pollination (**A′**), but no visitor species discriminated between SI and SM flowers under enclosed conditions (**A–D**). All visitor species spent more time (**E′–H**) and touched more stamens (**I′–L**) in SI flowers than in control SM flowers. Visitors removed more nectar from SI flowers, resulting in less nectar remaining per flower (**M′–P**). Pollen removal by *A. cerana* was lower from SI than from SM flowers (**Q′, Q**), but higher in the other three visitor species (**R–T**). Compared to SM flowers, pollen transfer efficiency was significantly decreased in SI flowers (**U′, U, V, X**), although it did not differ in *Meliscaeva* spec. (**W**). Sample sizes (n) are given in brackets.

was loaded with fewer grains than in SM flowers (just as under outdoor conditions; above), while the anthophorid bees and the flies under enclosed conditions removed more grains from SI flowers than from SM flowers (*Figure 4Q′–T*).

## Effects of mobile stamens on pollen receipt in experimental visits

After experimental single visits, both *A. cerana* (*Figure 5A*, Wald $\chi^2$ = 142.565, p<0.001) and the anthophorids (*Figure 5B*, Wald $\chi^2$ = 14.236, p<0.001) carried fewer pollen grains on their tongues after visiting SI flowers compared to SM flowers.

In three trials in which we held anesthetized bees between forceps and simulated visits to SM and SI flowers in different sequences ('Materials and methods'), the number of pollen grains deposited on stigmas by *A. cerana* (mean ± SE, *Figure 5C*) and the anthophorids (mean ± SE, *Figure 5D*) in trial 1 (SM + SM flower) was significantly higher than in trials 2 (SM flower + SI flower) and 3 (SI + SI flowers; Wald $\chi^2$ = 118.887, p<0.001 vs. Wald $\chi^2$ = 69.274, p<0.001, respectively). Moreover, pollen receipt by *A. cerana* and the anthophorids was significantly higher (p<0.001) in trial 2 (SM + SI) than in trial 3 (SI +SI), indicating that the stamens precisely place pollen grains on tongues, which then deposit them on stigmas during the insect's next flower visit.

## Effects of mobile stamens on pollen export and receipt in the field

Pollen export of SM and SI flowers was quantified with pollen-tracking experiments in which pollen of flowers of *B. julianae* and *B. jamesiana* was stained in situ with either eosin or aniline blue, and stigmas of all flowers in the vicinity (about 25–100 cm from the source plant) were checked for stained grains.

Of over 700 flowers of *B. julianae* whose stigmas we checked, 44 of 772 flowers received pollen from SM flowers and 14 of 733 flowers from SI flowers (*Supplementary file 3*), indicating that flowers with mobile stamens donated pollen to about 3.1× more flowers (44/772 = 0.057 vs. 14/733 = 0.019, G = 15.341, p<0.001). The mean number of pollen grains deposited per stigma in these trials was also higher from SM than SI flowers (0.16 ± 0.03 vs. 0.03 ± 0.01; Wald $\chi^2$ = 76.536, p<0.001). All four runs of this experiment showed a consistent pattern: more pollen grains from SM flowers were delivered to more flowers (*Figure 6*, *Figure 6—figure supplement 1*).

Within <50 cm from the dyed pollen source, 25 of 338 flowers received pollen from SM flowers, while only 10 of 395 flowers received pollen from SI flowers (G = 9.645, p=0.0019, *Supplementary file 3*). At distances of 50–100 cm, 19 of 434 flowers received pollen from SM flowers, while only 4 of 338 flowers received pollen from SI flowers (G = 7.439, p=0.0064, *Supplementary file 3*), and at distances >100 cm, only pollen from SM flowers was detected on stigmas (on 6 of 222 inspected flowers).

When we conducted the same experiment in *B. jamesiana* (*Figure 6—figure supplement 1*), 75 of 400 nearby flowers received pollen from SM flowers and 40 of 400 flowers from SI flowers, again indicating that mobile stamens were likely to donate pollen to more flowers (75/400 vs. 40/400, G = 12.612, p<0.001). The mean number of pollen grains deposited per stigma was also higher in SM than SI flowers (0.4 ± 0.05 vs. 0.2 ± 0.04; Wald $\chi^2$ = 22.95, p<0.001).

Within <25 cm from the dyed pollen source, 13 of 60 flowers received pollen from SM flowers, while 9 of 46 flowers received pollen from SI flowers (G = 0.07, p=0.791). At distances >25 cm, 7 of 60 flowers received pollen from SM donors, while only 1 of 56 flowers received pollen from SI flowers (G = 4.961, p=0.026, *Supplementary file 4*).

## Berberine content in *B. julianae* leaves, petals, pollen, and nectar

High-performance liquid chromatography (HPLC) of the berberine concentrations in *B. julianae* leaves, petals, pollen, and nectar indicated high berberine concentrations in leaves, petals, and pollen, while no berberine was detectable in the nectar (*Supplementary file 5*). That bees can taste the berberine

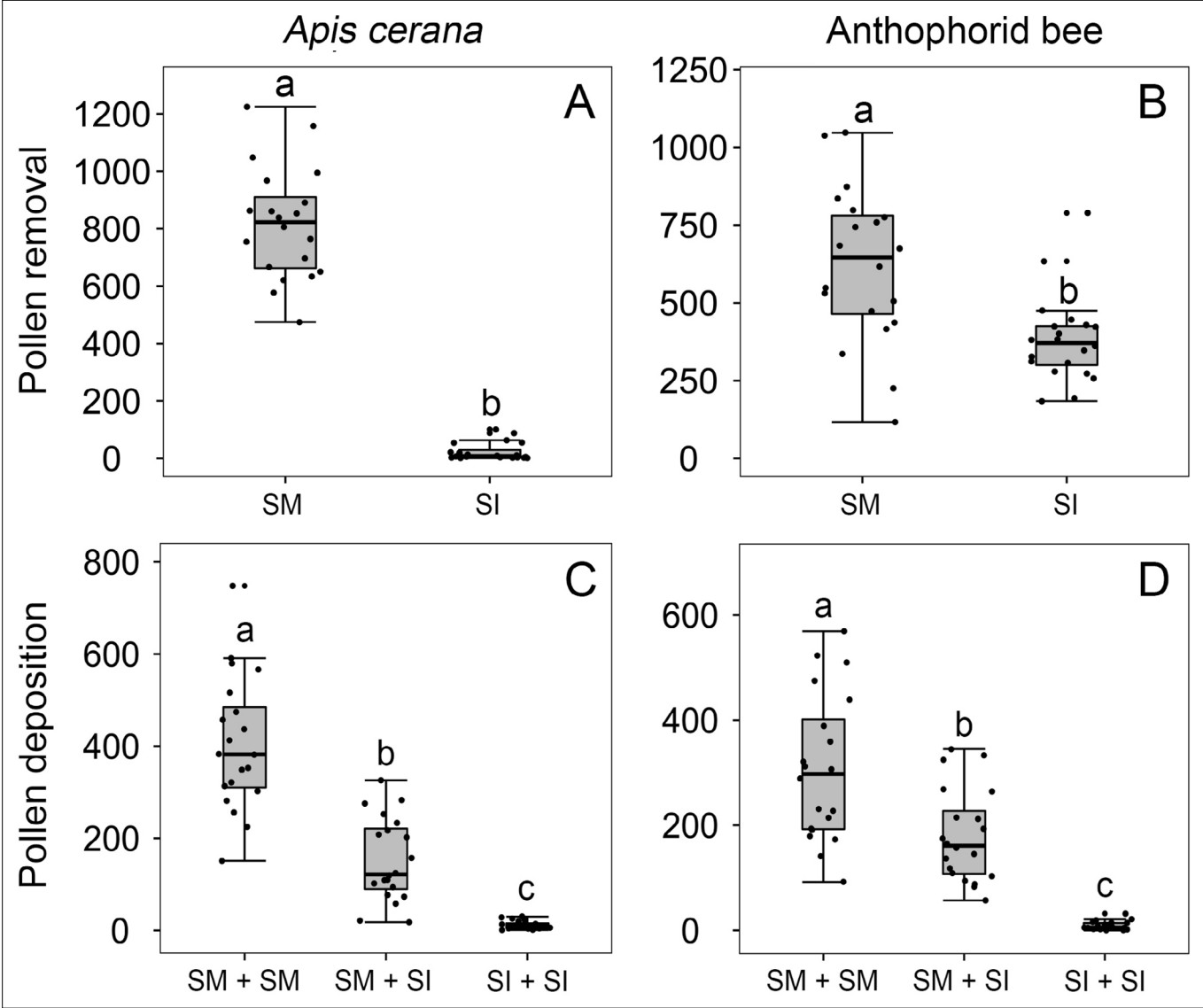

**Figure 5.** Effects of stamen forward-snapping on pollen placement on the pollinator body and pollen deposition on stigmas after single visits by *Apis cerana* (**A, C**) and the anthophorid bees (**B, D**). Numbers of pollen grains placed on bees' tongues during a single visit were significantly higher when stamens were mobile (SM) than when stamens were experimentally immobilized (SI) (**A, B**). Numbers of pollen grains deposited on the stigma of the second-visited flower (pollen recipient) during single visits by *A. cerana* (**C**) and the anthophorids (**D**) in three trials with the sequence being SM + SM flowers; SM + SI flowers; and SI + SI flowers. The box plots indicate the median (mid lines), the interquartile range (boxes), and 1.5× the interquartile range (whiskers). Different lowercase letters indicate significant differences among three trials.

The online version of this article includes the following figure supplement(s) for figure 5:

**Figure supplement 1.** Foraging behavior of *Apis cerana* and its effect on pollination in *Berberis julianae* flowers with experimentally immobilized stamens and controls in 2020 and 2021 including visits to floral arrays per flower (**A, B**), insect handling time per flower (**C, D**), and pollen removal (**E, F**), pollen deposition (**G, H**) and pollen transfer efficiency (**I, J**) by single visits.

in the pollen is suggested from the observation that individuals of all bee species used their front legs to remove pollen grains that stuck to their tongues.

## Discussion

Insect-induced movements of flower parts, including styles or stamens, have fascinated botanists since Linnaeus (1755) but their adaptive significance has been difficult to investigate because immobilization under field conditions has been difficult. As far as we are aware, this is the first study to directly study

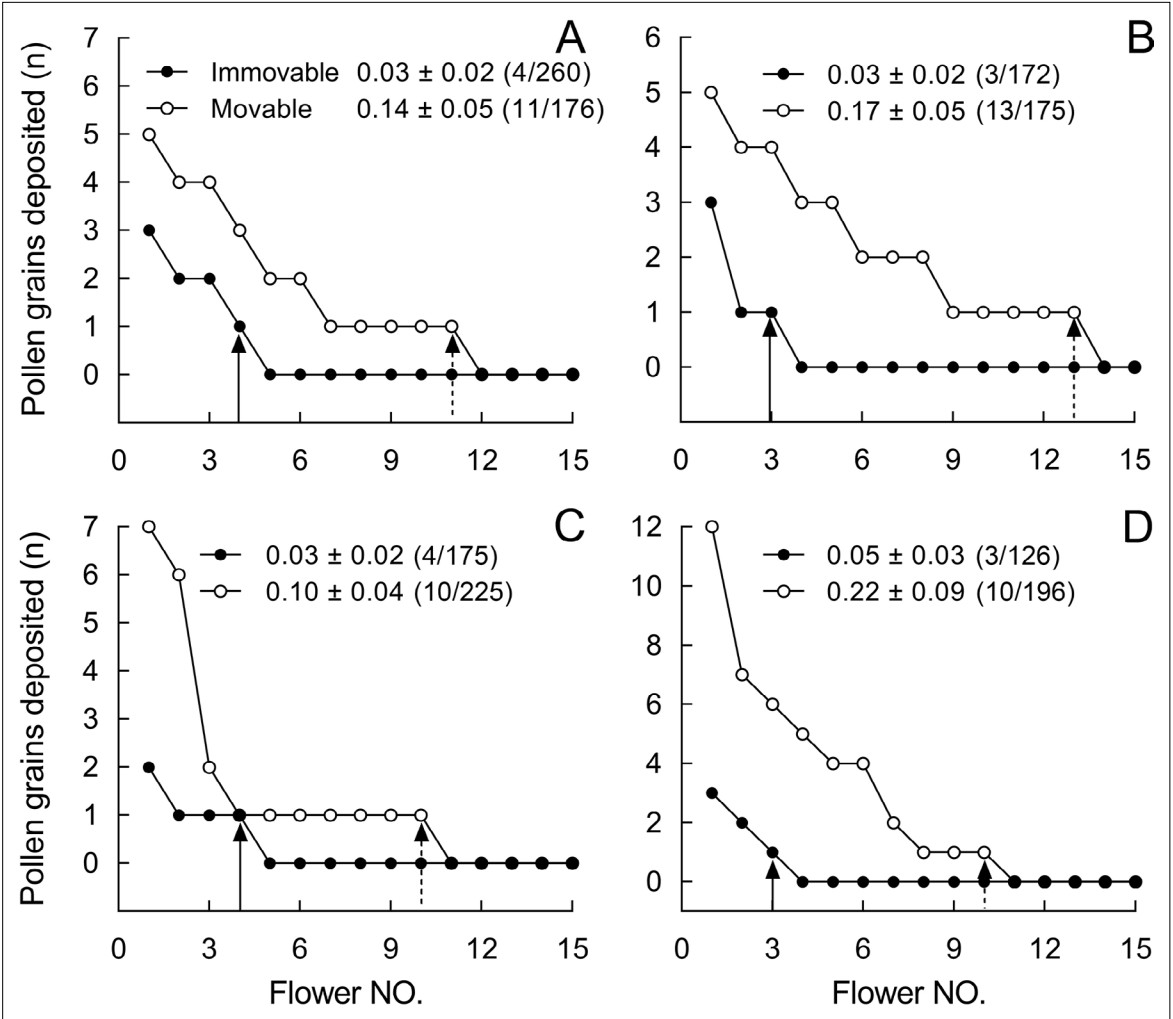

**Figure 6.** Number of stained pollen grains deposited on the stigmas from control flowers with mobile stamens (open circles) and from flowers with experimentally immobilized stamens (closed circles) in *Berberis julianae*. Mean pollen number and SEs (numbers of flowers with stained pollen deposition/total number of sampled flowers of pollen recipients) are given for treated and control flowers in each of four trials (**A–D**). Note that only 15 flowers are shown, although each pollen-tracking test sampled over 100 flowers to examine the effect of stamen movements on pollen dispersal; for example, of 260 pollen-recipient flowers, only 4 flowers received pollen grains from stained stamens-immobilized (SI) flowers, that is, (4/260) in (**A**).

The online version of this article includes the following figure supplement(s) for figure 6:

**Figure supplement 1.** Number of stained pollen grains deposited on the stigmas from control flowers with mobile stamen (open circles) and from flowers with experimentally immobilized stamens (closed circles) in *Berberis jamesiana*.

visitor behavior and pollen import/export in flowers with experimentally immobilized stamens, using our discovery that an alcohol solution blocks stamen movement in *Berberis* and *Mahonia* flowers. We developed a simple protocol of immersion of pedicels in alcohol and then carefully inserted flowers with immobilized stamens in racemes with the same number of natural flowers (directly in the field) or exposed them to visitors in experimental arrays (under enclosed conditions). This allowed us to test three hypotheses explaining the adaptive value of touch-sensitive stamens in Berberidaceae: visitor filtering; making pollinators leave more quickly, thus reducing nectar costs per pollen grain movement; and precise metering out of few pollen grains per visitor, enhancing pollen export. We found support for the last two, but not for the filtering hypothesis, since flies and bees reacted similarly to stamen bending. In the following, we discuss these results in more detail.

To maximize pollen dispersal, the number of grains removed by each visitor should be limited so as to heighten the probability of pollen reaching different conspecific stigmas (*Harder and Thomson, 1989*), and where possible, visitors should be 'paid' by replenishable nectar rather than pollen (*Westerkamp, 1996*). In *Berberis*, such metering out of pollen is achieved by the individual stamens only

bending forward once their adaxial filament base is touched by an insect's tongue probing the nearby nectaries. Further protection of pollen grains from exploitation by pollen thieves or inefficient vectors is sometimes achieved by chemical defense (*Palmer-Young et al., 2019*; *Wang et al., 2019*), and we therefore examined whether the pollen grains of *Berberis* contain berberine, an alkaloid with antifeedant activity against herbivores and pests (*Schmeller et al., 1997*; *Manosalva et al., 2019*). This is indeed the case (*Supplementary file 5*), and in *B. julianae*, we observed bees cleaning off and discarding pollen grains that stuck to their tongues with their front legs. The syrphid flies, however, fed on the pollen despite its berberine content. There is therefore no support for a chemical defense against pollen thieves (here pollen-feeding flies) and some support for bees being discouraged from collecting *Berberis* pollen as food for their larvae. Importantly, the syrphids visiting *B. julianae* are only conditional pollen thieves (*Hargreaves et al., 2009*) since the species is self-compatible and even self-pollen deposited by flies or by stamens hitting the flower's own stigma contributes to reproductive insurance.

The quantitative effect of stamen bending on the duration of bee visits per flower was large. Thus, in stamen-bending flowers, the bees typically stayed for five seconds and triggered 2–3 stamens per flower per visit (*Figure 4*), a similar number as in *Berberis thunbergii* in North America, where the main visitors also are medium-size bees of in the genus *Apis* and the family Anthophoridae (*Lebuhn and Anderson, 1994*). In stamen-immobilized flowers, bees stayed about 3× longer (on average 14.37 ± 1.53 s) and flies about 2× longer (on average 54.38 ± 7.53 s), and both visitor types therefore removed more nectar. The impact of longer stays on pollen removal, however, differed between the most common visitor, *A. cerana*, and the other three visitors: When immobilized stamens no longer smeared pollen grains on its tongue or face, *A. cerana* carried away fewer grains, while the pollen-feeding syrphid flies and the two anthophorid bees removed more grains, in the case of the flies because they ate more pollen and in the case of the anthophorids because they passively touched more open anthers, the longer they stayed. The quicker leaving of flowers not only reduced nectar costs/pollen grain transported but also greatly increased male reproductive success by dispensing more pollen to more recipients. This is evident from pollen dispersal distances in SM and SI flowers of *B. julianae* (*Figure 6*, *Supplementary file 3*) and *B. jamesiana* (*Figure 6—figure supplement 1*, *Supplementary file 4*), and from the proportions of pollen recipients that were reduced by 62.03% [=1 – (14/733 ÷ 50/994)] and 41.18% [=1–{(9+1)/(46+56) ÷ (13+7)/(60+60)}] in the two species.

While we found no support for touch-sensitive stamens filtering floral visitor types – all visitors left flowers more quickly after being hit by stamens– other studies on species with stamens triggered by flower visitors have found strong support for touch-sensitive stamens filtering out inefficient pollinators. Thus, in species of *Opuntia*, the stamens touched by bees move toward the flower center, creating a dense layer that effectively excludes generalist bees, while three specialized bee species are able to access the pollen (*Schlindwein and Wittmann, 1997*). And in *Meliosma tenuis*, only nectar-seeking bumblebee drones are able to trigger the stamens and are then loaded with pollen (which males do not collect as larval food), while other insects are unable to access the pollen (*Wong Sato and Kato, 2018*).

Recent reviews of the adaptive significance of the movement of floral parts (including both styles and stamens) have distinguished four types of stamen movement: (1) slow movement triggered by a visitor, (2) quick or even explosive movement (sometimes via a catapult mechanism) triggered by a visitor, (3) simultaneous slow movement occurring without triggering by a visitor, and (4) a 'cascade' or staggered movement in which one stamen moves after the other, regardless of triggering (*Ren, 2010*; *Ruan and da Silva, 2011*; *Armbruster et al., 2014*). The Berberidaceae, a family of some 700 species, exhibit aspects of two of these categories, namely, the staggered movement and the quick triggered movement, although not involving the sudden release of a built-up tension as, for instance, in *Kalmia* (*Switzer et al., 2018*) in which each filament is arched backward in a petal pocket. Experimental work on the costs and benefits all these types of stamen movements, which occur in hundreds of flowering plants, is still in its infancy, however, partly because it is so difficult to manipulate floral movements under field conditions.

## Conclusion

Even though botanists have speculated about the adaptive value of the visitor-triggered forcefully forward-snapping stamens of *Berberis* since 1755 (*Linnaeus, 1755*), this is the first experimental

investigation of how this trait impacts the flowers' pollen export and receipt. Our results demonstrate surprisingly large effects of stamen bending on pollen export (involving both quantity and distance) and nectaring times (involving lower nectar costs per pollen grain transfer) and reveal another mechanism by which plants meter out their pollen.

## Materials and methods

### Plant and insect study species

During each March between 2019 and 2022, we studied a natural population of *B. julianae* C.K.Schneider in a field located at 29°52′26″N, 105°30′32″E, 427 m above sea level, about 50 km southeast of Anyue County, Sichuan Province, China. Experiments to evaluate the effect of stamen movements on pollen dispersal were also carried out in a natural population of *B. jamesiana* Forrest & W.W.Smith at Shangri-La Alpine Botanical Garden (27°54′05″N, 99°38′17″E, 3300–3350 m above sea level), Yunnan Province, Southwestern China. We also studied planted populations of *B. forrestii* Ahrendt at the Shangri-La field station and of *M. bealei* (Fortune) Carrière in the Wuhan Botanical Garden (30°33′2″N, 114°25′48″E, 23 m above sea level) in Hubei Province to test the effects of the alcohol treatment on stamen mobility. Our *Berberis* and *Mahonia* taxonomy follows the Flora of China (*Ying et al., 2011*). Herbarium vouchers of each species have been deposited in the herbarium of Central China Normal University (CCNU). All species are hermaphroditic perennial shrubs with clusters of 9–25 yellow flowers produced between March and May, depending on species. Individual flowers last for 3–5 days, and each anther dehisces upward by two valves exposing the pollen grains (*Figure 1*). The bottle-shaped pistils have one ovary containing 2–4 ovules with a discoid stigma with a peripheral ring of papillae (*Figure 1E*).

Insect visitors were observed, in some cases filmed, captured with insect nets in the field, and preserved for later identification by insect taxonomists.

### Alcohol as an inhibitor of the stamen movement, and tests for confounding the effects of the alcohol treatment on visitor behavior

When we immersed floral pedicels in a solution of 75% alcohol for 40 min, all stamen movement was blocked (*Videos 1* and 3 of *B. julianae* and *Video 2* of *M. bealei*). To test whether any lingering alcohol scent affected visitor behavior in alcohol-treated flowers, we set up arrays with different types of flowers as follows. We first bagged >20 flowers on different individuals of *B. julianae* before they opened. Once open, 18 flowers were gently cut off and subjected to one of the three treatments: (1) stamen-mobile (SM) flowers: six flowers without any treatment; (2) stamen-immobilized (SI) flowers: six flowers whose pedicels (ca. 10 mm long) were immersed in 75% alcohol for about 40 min; and (3) six natural flowers in a fixed position above alcohol (SMA flowers). For this, the pedicels of freshly opened flowers were inserted into 30-mm-long Eppendorf microcentrifuge tubes that contained

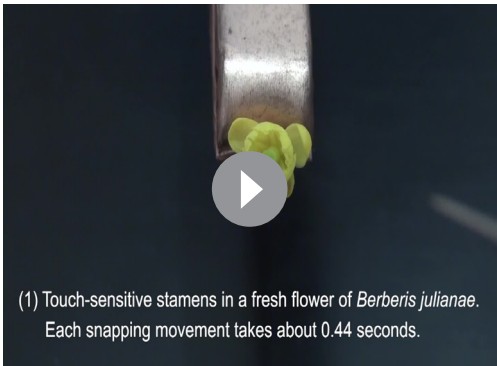

**Video 1.** Stamens of *Berberis julianae* become touch-insensitive after the flower pedicels had been immersed in 75% alcohol for 35 min.

https://elifesciences.org/articles/81449/figures#video1

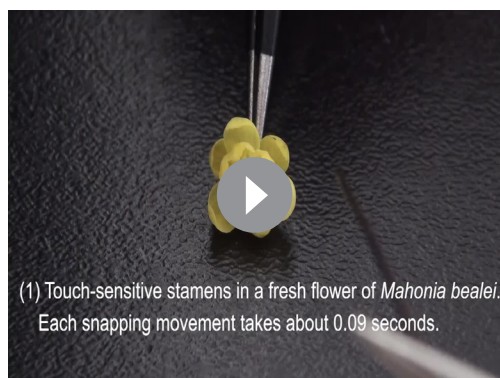

**Video 2.** Stamens of *Mahonia bealei* become touch-insensitive after the flower pedicels had been immersed in 75% alcohol for 30 min.

https://elifesciences.org/articles/81449/figures#video2

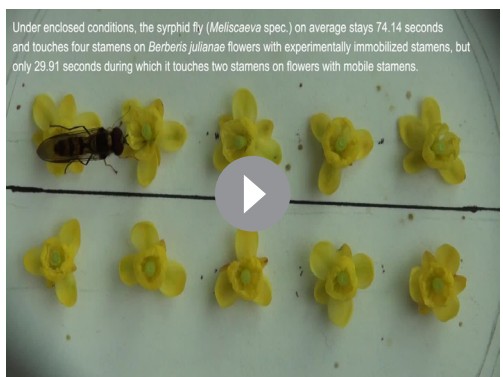

Under enclosed conditions, the syrphid fly (*Meliscaeva* spec.) on average stays 74.14 seconds and touches four stamens on *Berberis julianae* flowers with experimentally immobilized stamens, but only 29.91 seconds during which it touches two stamens on flowers with mobile stamens.

**Video 3.** Under enclosed conditions (Figure 1—figure supplement 1D), individual syrphid flies (*Meliscaeva* spec.) took up nectar from a *Berberis julianae* flower with immobilized stamens for much longer, giving them time to touch four stamens, while leaving more quickly and touching only two stamens in a control flower with mobile stamens.

https://elifesciences.org/articles/81449/figures#video3

10 µL alcohol, such that they did not touch the alcohol (*Figure 1—figure supplement 1A*). All 18 flowers were then fixed in position by inserting them into small holes on the surface of a paper box covered by a clean glass cup (as shown in *Figure 1—figure supplement 1*). We then placed freshly caught *Berberis* visitors inside the glass cup so they could interact with the enclosed floral arrays and recorded visitation rates (visits per flower per 10 min) and handling times.

To further examine the possible effects of the alcohol treatment versus the loss of stamen movements on the duration of insect visits, we set up additional floral arrays consisting of three types of flowers: six SM flowers, six alcohol-treated SI flowers, and six FD flowers, the latter being flowers in which all six filaments were damaged with clean forceps so that the stamens could not move but petal nectaries and the anther sacs were still there (*Figure 1—figure supplement 1C*). The experimental procedure for these arrays was the same as above (*Figure 1—figure supplement 1B*).

## Fruit and seed set after self-pollination vs. cross-pollination, and contribution of stamen forward-snapping to selfing

To test whether *B. julianae* is self-compatible and whether fruit or seed production are pollinator-limited, in 2020 and 2021, 80 flowers from 12 individuals were subjected to the following four pollination treatments. (1) Control: 20 randomly chosen flowers from 12 individuals were open pollinated, while the remaining 60 flowers were bagged and subjected to one of three treatments: (a) automatic self-pollination: flowers not manipulated; (b) self-pollination: flowers hand-pollinated with pollen mixtures collected from flowers of the same individual; (c) cross-pollination: flowers hand-pollinated with pollen mixtures from multiple flowers of individuals at least 20 m away. Flowers were then bagged with mesh until the petals dropped 1 week later. Fruits were harvested 3 months later and seeds and undeveloped ovules per fruit counted. Fruit set was calculated as fruit number divided by flower number in each treatment. Seed set was calculated as seed number per fruit divided by total number of seeds and undeveloped ovules. Aborted fruits were not included.

To examine intra-flower selfing induced by the stamen movement, we counted pollen grains deposited per stigma under a light microscope in 16 bagged flowers in which we had triggered one stamen by a needle.

## Effect of stamen bending on visitor behavior and pollen export and import

Flowers of *B. julianae* were visited by several species of bees and two species of syrphid flies ('Results'). To examine the effects of stamen bending on foraging behavior and pollen transfer, floral visitors were allowed to interact with the above-described flower arrays (section 'Alcohol as an inhibitor of the stamen movement, and tests for confounding the effects of the alcohol treatment on visitor behavior') in the field (*Figure 1*) and under enclosed conditions (*Figure 1—figure supplement 1D*). Under enclosed conditions, we compared visitation rates, handling times, numbers of stamens touched, nectar volume remaining, pollen removal, and pollen receipt on the stigma after single visits of the four species of visitors. We also quantified pollen transfer efficiency after single visits by harvesting flowers that had been visited a single time in the field and then counting pollen grains on their stigmas as well as pollen grains remaining in their stamens. To quantify pollen loads placed on insect bodies and stigmas of next-visited conspecific flowers, we held captured and anesthetized bees between

forceps and made their tongues contact the filament bases in SI or SM flowers. Using this method, we carried out three trials with SM and SI flowers: trial 1, SM + SM; trial 2, SM + SI; and trial 3, SI + SI flowers.

## Pollen-tracking experiments to quantify the effects of stamen forward-snapping on pollen export distances

To quantify pollen export from SM and SI flowers, we conducted four pollen-tracking experiments each in *B. julianae* in March 2022 and *B. jamesiana* in May 2021. Each trial was conducted on a sunny day and involved 60 flowers from 3 to 5 individuals whose pollen was stained as the pollen donor: 30 flowers were alcohol-treated (SI flowers) and the remaining 30 flowers were SM flowers. The manipulated flowers were carefully inserted among natural flowers on densely blooming racemes. The stigmas of >100 flowers (126–260) on the same and on different individuals at varying distances from the donor were then examined for dyed pollen grains, these were counted, and the straight-line distance from donor to recipient recorded with a meter rule.

## Statistical analyses

Visitation rates of visitor species in the field (not normally distributed) were examined with nonparametric Kruskal–Wallis tests. To compare visitor behavior on SM, SMA, FD, and SI flowers, we performed a generalized linear model (GLM) analysis with normal distributions and identity-link function to test for differences in visitation rates, handling time, and pollen transfer efficiency (log10-transformed) of the four main visitor species. Nectar volumes remaining after different visitors had visited were examined with a nonparametric Kruskal–Wallis test. To compare the number of stamens touched in different flowers, pollen removal, and pollen deposition by each visitor species, and pollen export and import, we performed GLM analyses with a Poisson distributions and loglinear-link function. Floral traits among species were compared under a GLM with a normal distribution and identity-link function, while for numbers of pollen ovules per flower we used a Poisson distribution and loglinear-link function (*Supplementary file 1*). Data of pistil height and stamen length were log10-transformed to achieve normal distribution. We used the G-test of independence to test whether stained pollen from SM or SI donors differs in the distances to which it is dispersed to (*McDonald, 2014*). A GLM with binomial distribution and logistic-link function was used to detect the effects of the selfing and outcrossing on fruit set and seed set (with fruit/seed number as event variable, total treated flower/ovule number as trial variable, and different treatments as factors). The GLM analyses were performed in SPSS 19.0 (IBM, Armonk, NY).

## Acknowledgements

We thank lab members Q-M Quan, X-W Lv, and Z-X Tian for field assistance and Z-Y Tong, and Y-Z Xiong for methodological and statistical advice; Dr. Huan-Li Xu, Department of Entomology, China Agricultural University, Beijing, for the identification of flies and bees; Dr. Antti Haarto, Zoological Museum, University of Turku, Finland, for the identification of *Meliscaeva*; Dr. Michael Orr, Institute of Zoology, Chinese Academy of Sciences, Beijing, for the identification of *Habropoda* cf. *sichuanensis*; Dr. Chih-Chieh Yu for confirming our plant identifications; Z-D Fang and staff of Shangri-La Botanical Garden for logistical support; Sarah Corbet, Steven Johnson, and Nathan Muchhala for providing helpful suggestions on early versions of this manuscript; and the reviewers Bernhard Schmid, Zong-Xin Ren, and Felipe Yon for their constructive comments. This work was supported by the National Natural Science Foundation of China (grant nos. 31730012 and 32030071) and Fundamental Research Funds for the Central Universities (no. CCNU22LJ003) to S-QH.

## Additional information

### Funding

| Funder | Grant reference number | Author |
|---|---|---|
| National Natural Science Foundation of China | grants no. 31730012 and 32030071 | Shuang-Quan Huang |
| Fundamental Research Funds for the Central Universities | no. CCNU22LJ003 | Shuang-Quan Huang |

The funders had no role in study design, data collection and interpretation, or the decision to submit the work for publication.

### Author contributions

Deng-Fei Li, Resources, Formal analysis, Investigation, Methodology, Writing – original draft; Wen-Long Han, Methodology; Susanne S Renner, Writing – original draft, Writing – review and editing; Shuang-Quan Huang, Conceptualization, Resources, Formal analysis, Supervision, Funding acquisition, Visualization, Writing – original draft, Project administration, Writing – review and editing

### Author ORCIDs

Susanne S Renner http://orcid.org/0000-0003-3704-0703
Shuang-Quan Huang http://orcid.org/0000-0003-4540-1935

### Decision letter and Author response

Decision letter https://doi.org/10.7554/eLife.81449.sa1
Author response https://doi.org/10.7554/eLife.81449.sa2

## Additional files

### Supplementary files

• Supplementary file 1. Floral traits (mean ± SE) and duration of stamen movements in *Berberis jamesiana*, *B. julianae*, *B. forrestii*, and *Mahonia bealei*.

• Supplementary file 2. Foraging behaviors of two types of major insect visitors to *Berberis julianae*. Behaviors of the bees and flies visiting *Berberis julianae*, focusing on visitation rates (visits per flower per hour, mean ± SE), visits per flower, handling time, number of stamens touched, nectar volume remaining per flower, pollen grains removed, pollen grains deposited, and pollen transfer efficiency.

• Supplementary file 3. G-test of independence confirming that *Berberis julianae* flowers with mobile stamens (SM flowers) donated pollen to more recipient flowers at three distance classes than did flowers with experimentally immobilized stamens (SI flowers).

• Supplementary file 4. G-test of independence confirming that *Berberis jamesiana* flowers with mobile stamens (SM flowers) donated pollen to more recipient flowers at two distance classes than did flowers with experimentally immobilized stamens (SI flowers).

• Supplementary file 5. Concentrations of berberine (mean ± SE) in different tissues of *Berberis julianae* under generalized linear models.

• MDAR checklist

### Data availability

All data are shown in the main text and the tables at the end of the main text.

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

## Appendix 1

### Measurements of floral traits in *Berberis* and *Mahonia* species

Flower size, length of the pedicel, and other flower traits were measured with a digital caliper to 0.01 mm (*Supplementary file 1*). To estimate nectar volume per flower of *B. julianae*, clean 10 µL microliter syringes (Agilent Technologies Inc, USA) were used to extract nectar drops from bagged flowers. To count pollen grains and ovules per flower, we randomly collected one of six anthers from virgin flowers that had been bagged as buds with fine-mesh cotton bags to exclude visitors. Each selected anther was placed on a microscope slide and then squashed under a coverslip. All pollen grains were counted under a microscope, and the number in one anther was multiplied by six to obtain the pollen grain number per flower. Meanwhile, ovules of sampled flowers were also counted. Sample sizes are shown in *Supplementary file 1*.

### Testing protocols for best alcohol treatments to block the stamen movement

To find the minimum time required to completely block stamen movements in each of the three species of *Berberis*, we checked the response of stamen movements in a time series of alcohol-treated flowers. Pedicels were gently cut off and the bases (about 10 mm long) immersed in 75% alcohol. Every 5 min, 8–10 flowers were checked by touching the filaments of each flower with a dissecting needle to identify whether the stamens remain mobile. For *B. julianae*, we tested 64 flowers, for *B. jamesiana* 100, and for *B. forrestii* 80.

### Floral visitors and effect of stamen bending on their behavior and pollen export and import

Visitation rates (visits/flower/hour) of the different visitors to *B. julianae* were obtained on sunny days in March 2019, 2020, and 2021. Visitor were observed on fully flowering shrubs with over 100 open flowers, and each observation period lasted 1 hr, during 9:00–12:00 and 12:30–16:30. The sex and foraging behavior of each visitor were recorded, such a feeding on nectar or pollen or groomed pollen grains into pollen loads. Total observation times in the 3 years are shown in *Supplementary file 2*.

To see whether floral visitors discriminated between SI and SM flowers, we compared visitation rates of the Asian honeybee *A. cerana* under open pollination and under enclosed conditions. The other insect species were only studied under enclosed conditions (illustrated in *Figure 1—figure supplement 1*).

In March 2020, 72 flowers on different individuals of *B. julianae* were randomly bagged with mesh nets, before they opened. Also, 12 SI and 12 SM flowers were observed each day. When flowers were beginning to open, we gently cut off 36 flowers (12 flowers × 3 days) at the base of the pedicels and immersed the pedicels in 75% alcohol. All stamens in each alcohol-treated flower became touch-insensitive in 40 min. These SI flowers were carefully inserted into racemes of flowering individuals with the other 36 SM flowers on these racemes as controls, allowing the bees to visit in open pollination conditions (*Figure 1D and H*). The number of flowers visited per hour was recorded during 09:00–12:00 and 13:00–17:00 for 3 days. Visitation rates were calculated as the number of flowers visited per hour divided by the number of observed flowers, that is, visits per flower per hour.

In March 2021, a total of 160 floral buds from five individuals were randomly bagged. As above, 80 newly opened flowers were alcohol-treated and the remaining 80 flowers were free of alcohol as SM flowers. On each observation day, we set up four arrays each having four SM and four SI flowers. The visitation rates were recorded during 09:30–12:30 and 13:00–17:00 for 5 days each with 16 SI and 16 SM flowers (*Figure 5—figure supplement 1A-B*).

To compare the handling time, pollen removal, and pollen transfer efficiency of the pollinator *A. cerana* in SM and SI flowers under open conditions, 100 flowers from different individuals of *B. julianae* were randomly bagged with mesh before the flowers opened in March 2020 and 2021. When the flowers were newly opened, 12 flowers per day were alcohol-treated for 5 days per year with fine weather. Four of the SI flowers were carefully inserted into each of three racemes with buds, but no open flowers (4 flowers × 3 racemes × 5 days = 60), and four bagged SM flowers from two racemes were uncovered per day (4 flowers × 2 racemes × 5 days = 40) for *A. cerana* visits. In the

five arrays, each of four flowers per day, we recorded handling times of *A. cerana* during each visit to one flower. To estimate pollen transfer efficiency, 20 SI and 20 SM flowers visited once by *A. cerana* were harvested immediately in the field. Six anthers and the stigma of each flower were collected. We counted pollen grains remaining within anthers per flower and deposited per stigma under a light microscope. Pollen removal per flower was calculated as the mean number of pollen grains (see 'Results') minus the remaining grains. Pollen transfer efficiency was calculated as pollen deposition divided by pollen removal. In both years, we obtained 20 pairs of pollen removal and deposition data for SM flowers, and 20 pairs for SI flowers. In March 2021, we also compared the number of stamens touched and the remaining nectar volume left by *A. cerana* in SM and SI flowers that it had it visited just once. As visitor switched to collect nectar from another nectary, their bodies turned. Therefore, counting the visitor's body turns allowed us to record the number of stamens touched by a visitor in SI flowers (*Video 3*).

To estimate nectar collection by different visitors, we measured the remaining nectar volume within each SM and SI flower after one visit using a clean 10 μL microliter syringe (Agilent Technologies Inc).

To compare the foraging behavior of bees and flies on SM and SI flowers under enclosed conditions (*Figure 1—figure supplement 1*), we set up floral arrays each consisting of five SM and five SI flowers. The flowers were fixed by inserting the pedicels into small holes on the surface of a paper box covered by a clear glass cover. We then allowed individual of the different insect species to freely interact with the floral array. We recorded the visitation rates (visits per flower per 10 min) and handling time of the visitor and the number of stamens touched by the visitor in each flower, measured the nectar volume left by the visitor, and assessed pollen removal and pollen deposition after one visit to calculate pollen transfer efficiency. It took four trials of 'interviews' (cafeteria experiments) over 2 days to yield 20 once-visited flowers for each insect species. Four individuals of each insect species were captured in the field and released within 2 hr after their visits to the SM and SI flowers.

## Pollen export/import after single visits

To estimate the effects of the stamen mobility on pollen placement on the pollinator and on stigmas, we simulated bee visits by to SI and SM flowers. In March 2021, we bagged 320+ flower buds from different individuals of *B. julianae*. When the flowers opened, we held anesthetized bees between forceps so that they probed the flower, pollen grains placed on the bee's tongue (and occasionally on the head) were pasted onto a piece of tape. The tape was attached to a slide and pollen grains placed on the bee's tongue in 20 SI and 20 SM flowers were then counted under a light microscope (*Figure 5A and B*). Using this method, we also carried out three trials with simulated visits to flowers in specific sequence: Trial 1, SM + SM flowers; trial 2, SM + SI; and trial 3, SI + SI flowers. Each trial was repeated 20 times. After each trial, the stigma of the second-visited flower (the pollen recipient) was collected to count the pollen grains deposited on its stigma (*Figure 5C and D*).

## Pollen-tracking experiments to quantify the effects of stamen forward-snapping on pollen export distances

To examine the effects of stamen movements on the fate of pollen, we conducted four trials of pollen-tracking experiments in a transplanted population of *B. jamesiana* at Shangri-La Alpine Botanical Garden, where the interference of stained pollen with the sexual reproduction of wild plants could be minimal. Trials were conducted on sunny days and involved 60 flowers from 3 to 5 individuals whose pollen was stained as pollen donors: 30 flowers were alcohol-treated (SI flowers) and the remaining 30 flowers were used as SM flowers. Pollen grains in SM or SI flowers were stained with eosin (stained red) or aniline blue (blue). The two dyes were alternately used between SM and SI flowers in four trials. The dyes dried within 5–10 min, and all pollen-stained flowers were taken to the field and carefully inserted into racemes of two flowering individuals along the roadside for pollinators to visit. To track pollen flow, the two clusters containing either 30 SM or SI flowers were arranged separately (about 100 cm apart) on the flowering branches. Flowers of each cluster were within a 40 cm × 40 cm square on four erect racemes. Previous studies indicated that pollen flow mediated by generalist insects usually occurs within meters, with a highly leptokurtic distribution (*Williams, 2001*). One day (24 hr) later, given that *A. cerana* visits were infrequent to *B. jamesiana* in May 2021, stigmas of 100 flowers from nearby racemes were collected. Dyed pollen grains deposited on each stigma were counted under a stereomicroscope.

To compare the distance of pollen dispersal from SM and SI flowers, we conducted two trials of pollen-tracking experiments to compare distance of pollen dispersal between SM and SI flowers of *B. jamesiana*. Each trial was the same as above. At 17:30 on the day of the experiment, stigmas of all flowers from nearby racemes were collected and we noted the straight-line distance within 25 cm and over 25 cm from the racemes with the sampled flowers to the racemes with pollen-stained SM or SI flowers. Dyed pollen grains deposited on each stigma were counted under a stereomicroscope.

## Measurement of chemical defense in pollen and nectar

To examine whether berberine is present in *Berberis* pollen and nectar, we collected leaves, petals, pollen, and nectar from 10 plants of *B. julianae*. Leaves, petals, and pollen grains were dried using an oven at 65°C, while the nectar was stored at –20°C before chromatographic analysis. The berberine content of the different samples was analyzed using HPLC. To extract berberine from leaves, a 0.1 g leaf sample was weighed with a balance (Sartorius BAS124S) and transferred to a 2 mL microcentrifuge tube. A steel bead was added to the tube. The leaves were ground using a tissue homogenizer (Tissuelyser, QIAGEN) at 30 Hz for 10 min, and then the leaf sample was transferred to a vial. Enzymolysis of the leaf sample was conducted by adding 2 mL pure water, 1 µL dilute sulfuric acid, and 6 mg cellulose at 50°C in a magnetic stirrer for 3 hr. The leaf sample was extracted for 2.5 hr in 5 mL pure water, and then transferred to a 5 mL microcentrifuge tube. Following centrifugation at 8000 rpm for 10 min (Centrifuge 5430R, Eppendorf), the supernatant was transferred to a vial. The volume of the sample was recorded. Berberine in petals, nectar, and pollen grains was measured similarly.

Components were separated using an Acquity HPLC BEH C18 column (50 × 2.1 mm, 1.7 µm) (Waters, Milford, MA) set at 30°C, and the injection volume was 20 µL. All aspects of system operation and data acquisition were controlled by software (Agligent 1100 series) at the Center of Analysis and Test of Wuhan University, Wuhan, China. The mobile phase was acetonitrile–0.3% phosphoric acid–pure water (35:5:60); flow rate: 1.0 mL/min; determination wavelength: 346 nm; the detection signal was diode array detector (DAD). The berberine concentrations in the different samples were compared using a GLM with normal distribution and identity-link function.

