## [Editor Report]

With a series of manipulative experiments using four plant species with stamens that can snap toward the stigma if touched at the base, the authors provide compelling evidence that pollinators stay longer yet export less pollen to recipient flowers when stamens are immobilized by alcohol application. This is a landmark study on the functional consequences and adaptive significance of a phenomenon scattered throughout the angiosperm clade.

---

## [Decision Letter]

**Decision letter after peer review:**

Thank you for submitting your article "Mobile stamens enhance pollen dispersal by scaring floral visitors away" for consideration by *eLife*. Your article has been reviewed by 4 peer reviewers, including Bernhard Schmid as Reviewing Editor and Reviewer #1, and the evaluation has been overseen Jürgen Kleine-Vehn as the Senior Editor. The following individuals involved in the review of your submission have agreed to reveal their identity: Zong-Xin Ren (Reviewer #3); Felipe Yon (Reviewer #4).

Essential revisions:

1) The different experiments need to be described in more detail both with regard to the design, procedure, measurements, and analyses. Provide more biological description of the receptivity, fertility, perception, day-cycle, and snapping capacity of the flowers over the 5 days that they are assumed to be opened for all species, including potential auto-deposition by spontaneous snapping.

2) Of the four hypotheses the first two have been tested with good experimental designs and statistical power (pollen export from mobile vs. immobile stamens). However, this was not the case for the other two (pollinator filtering, presence or effect of berberine in pollen on pollinators). This should be acknowledged and discussed throughout the manuscript.

3) Check that the entomology is correct, including species identity and natural history. Also, be careful regarding the interpretation of pollen removal as indicative of berberine content in pollen.

4) The Discussion should be improved by better synthesizing the different experiments into a general narrative and referring more extensively to general theory and other work on pollination aspects considered in this paper. Also, point out limitations and what would have to be done next to get a more comprehensive and mechanistic understanding of the function of mobile stamens (including pollinator behavior and niche separation as well as pollination results such as increased outcrossing rate or distance).

*Reviewer #1 (Recommendations for the authors):*

Besides the main results whether mobile stamens could "filter" visits according to pollinator species identity could not be confirmed and whether berberine affects pollinator behavior could only weakly be inferred from observation of pollen removal from bee tongues. It also remains to be shown whether mobile stamens additionally have a function in ensuring auto-deposition of pollen on stigmas for selfing and whether they increase seed production and offspring fitness of the plant.

While the experiments carried out are fine as far as they go, they are somewhat haphazardly planned using different species and places to test different partial questions relevant to the whole story. There is also some intrinsic confounding between different possible functions of mobile stamens, two mentioned by the authors (shorter visit time and more precise placement of pollen on insect body) and additionally the one of auto-deposition of pollen for selfing.

The experiments should be described in some more detail and using more precise wording. For example, there are places (e.g. line 257 and lines 295-299) where it is not clear if mobile vs. immobile stamen are compared on the side of the plant from which pollen is exported or on the side of the plant to which pollen is deposited (for example the latter is said in the legend to Figure 5, but from the text, the opposite seems to be the case). It is also not clear if self-pollen reception was tested in the absence or the presence of insects. Would stamens spontaneously snap in older flowers to ensure selfing in case they had not been visited?

The Discussion section is rather short, reflecting the relatively simple but interesting and novel results. Nevertheless, it seems there must be other literature about the effects of mobile stamens or at least the mechanism could be compared with other mechanisms assumed to reduce ineffective insect pollination in flowering plants or potentially enhance auto-deposition of pollen for selfing (relevant also because there was no difference in pollination success between hand-pollinating selfing and outcrossing treatments). Furthermore, the Discussion could point out where to go from the present "natural history" to more comprehensive studies in the model system. Finally, for this type of study, it would be useful to briefly discuss limitations.

Generally, the manuscript is well written and the analyses of the simple experiments are fine as far as it can be judged based on the brief descriptions.

*Reviewer #2 (Recommendations for the authors):*

This is an interesting report on the functional consequences and adaptive significance of a phenomenon scattered throughout the angiosperm clade: "explosive" snapping movement of stamens in response to being touched. Using four species of Berberidaceae, this study provides the first experimental evidence that rapid stamen movement and active pollen placement increase pollen dispersal to stigmas of other flowers. There is also evidence that the stamen-movement system results in pollen placement being more accurate with respect to sites of stigma contact and pollen being dispensed to a larger number of pollinators. These are all valuable insights into how natural selection has favored the evolution of complex floral behavior.

The writing is reasonably clear for the most part, although the sentence structure could be improved in many places. Analyses appear to be appropriate (although more information explaining and justifying the statistical approaches would be desirable). The results and conclusions appear to be valid.

Specific comments

1. Graphic-abstract caption and in main abstract: "Stamens only snap forward if their filament basis is touched by an insect tongue" implies that artificial stimulation (e.g. with a toothpick) doesn't trigger stamen movement. I don't think you mean this, and if not, rephrase.

2. Introduction line 5. "intended" must be in quotes or, better, not used at all.

3. Introduction, 1st paragraph. "Flowers are therefore under selection to 'pay' visitors as much as possible by nectar" is potentially misleading. Heinrich and Raven (Science 1972) argued that nectar production/standing crop would not be maximized, but rather optimized to promote pollinator movement.

4. Introduction, 2nd paragraph. I think you mean "base", not "basis".

5. Materials and methods a, paragraph 2. The preferred word is "pinned" not "needled."

6. Materials and methods c, 1st sentence. Don't use both passive voice ("were immersed") and active voice ("this blocked") in the same sentence. Change to, e.g., "all stamen movement was blocked…"

7. Results a, last line. "selfed" is ambiguous. I think you mean "manually self-pollinated".

8. Results f (and Statistical Methods section). I was unclear as to what assumptions and tests were used to assess the significance of differences in proportions of flowers receiving dyed pollen, e.g. "Of 400 nearby flowers in B. jamesiana, 75 received pollen from SM flowers and 40 from SI flowers, again indicating that mobile stamens donated pollen to more flowers (75/400 vs 40/400, G = 27.81, p < 0.001)." G-test doesn't explain the model or the assumptions, just implies log-likelihood. Is this assuming Normal or Poisson distributions or some nonparametric goodness of fit? The first doesn't seem appropriate and the latter yields a P-value slightly > 0.001 by my quick calculation. This needs a better explanation in the Method section. Because differential dispersal of pollen by motile vs. immobilized stamens is, arguably, the single most important result, its statistical validity should be made crystal clear.

9. The Discussion would benefit from editing and tighter organization. The language could be improved, removing awkward constructions. The discussion should perhaps start with a brief, well-organized summary of the major results (possibly enumerated), followed by comparisons with other species and studies.

*Reviewer #3 (Recommendations for the authors):*

Pollen, containing male gametes of flowering plants, functions in two roles, reproduction for plants and food resources for many insects, especially bees and hoverflies. From the plant aspect, the high fitness should achieve by dispersal pollen grains by the vector to the receptive stigmas of conspecific flowers; for the floral visitor aspect, the high benefit should collect more pollen grains by visiting fewer flowers. This is the pollen dilemma in pollination ecology. Many complex floral traits (concealed pollen grains, pollen/stamen mimicry) or floral behavior (floral part movement, pollen packaging/dispensing) could be driven by solving the "pollen dilemma". In this study, Li et al., presented a well-written paper with well-designed experiments to test the behavior and function of stamen movement in Berberis. The clever experiment design by using alcohol to immobilize stamens made the control and manipulating experiments possible. They found after immobilized stamens, bees visited flowers longer, but removed and deposited fewer pollen grains lowering plant fitness. Therefore, stamen movement drove insects move among flowers, and removed and deposited more pollen grains to conspecific flowers. In my opinion, this is an excellent example to illustrate how "floral trait/behavior" evolves to solve the "pollen dilemma".

The authors concluded that "These results demonstrate another mechanism by which plants simultaneously meter out their pollen and reduce pollen theft." I partly agree with this. The evidence to support pollen meter out is very strong, but the evidence to support "reduce pollen theft" is not strong enough.

The authors asked four questions, "(1) Are Berberis flowers with mobile stamens visited by the same types of insects and at the same rates as flowers with immobilized stamens? (2) Do flowers with mobile stamens have higher maternal and/or paternal success than those with immobilized stamens? (3) Do mobile stamens contribute to pollinator filtering by eliciting different behaviors in different insect taxa? And (4) is berberine, an alkaloid with antifeedant activity against herbivores that are found in Berberis leaves, also present in Berberis pollen or nectar?", in my opinion, questions (1) and (2) were well addressed in this research. Question (3) was not well addressed by the experiments. Question (4) was descriptive, there is a potential association with questions (1) and (2) but not well explained and tested.

Another weakness is the discussion selection. The authors did not fully discuss the novelty and application of key findings to floral trait/behavior evolution in a larger sense. The order of points of discussion also needs to be re-organized following questions and main results. For example, only one sentence talked about floral visitor filtering related to question (3).

Using alcohol to immobilize stamens is very important, should expand more to explain how alcohol immobilizes stamen movement in Introduction and Discussion.

The time of stamen movement should compare with the flower handling time. Is the time stamen movement always shorter than the time of flower handling by insects?

Another limitation of this study is that we still don't know if such floral behavior drive insects to move among individual plant. If such behavior only drives insects to move among flowers within the same plant, it cannot increase the diversity of paternal donors. Since a plant of Berberis can have many many flowers, it is not easy to test plant fitness at an individual plant level. Therefore, it is worth testing the function of stamen movement at individual plant level in the future using a model species producing countable numbers of flowers.

1. I found Figure 1B is very confusing. I checked Anthophora waltoni, see the following two references, I feel the bee in Figure 1B is so different with A. waltoni in the references. I suggest you should let your entomologist double-check the identification.

Mohamed Shebl, Li Qiang, Victor H. Gonzalez "Nesting Behavior, Seasonality, and Host Plants of Anthophora waltoni Cockerell (Hymenoptera: Apidae: Anthophorini) in Yunnan, China," Journal of the Kansas Entomological Society, 87(4), 345-349

Zhenghua Xie, Mohamed A. Shebl, Dongdong Pan et al., " Synergistically positive effects of brick walls and farmlands on Anthophora waltoni populations ", Agricultural and Forest Entomology 22(4), pg. 328, (2020); doi:10.1111/afe.12384

2. This statement on Meliosma tenuis is not accurate, in the case of Meliosma tenuis, very few workers of bumblebees visit flowers, one potential reason is that because workers could not collect the explosively released pollen.

3. "Flowers are therefore under selection to 'pay' visitors as much as possible by nectar". Don't drop into the trap that "nectar is for the goodness of floral visitor", nectar production is selected for the plant's own fitness, but such fitness is realized by floral visitation.

4. "Replenishable nectar". However, not all plant species replenish nectar, see the following reference. Luo, E.Y., Ogilvie, J.E. and Thomson, J.D. (2014). Stimulation of flower nectar replenishment by removal: A survey of eleven animal-pollinated plant species. J. Pollin. Ecol., 12, 52-62.

*Reviewer #4 (Recommendations for the authors):*

The study brings attention to a long-standing and assumed function of snapping stamens in Berberis spp. It contains four objectives to which the methodological approach is standard and most are broadly supported in the literature, even though the suitability of some designs could be argued as suboptimal proof of concept for generalization of certain pollination conclusions. This works aims to prove that visitation doesn't differ when stamens are able or not of snapping, but that the snapping mechanism improves the pollen uptake and transfer to pollinators and thus improves fruit formation, while the signature alkaloid compound of Berberis julianae is not a detriment to the interaction.

The results support well the null hypothesis of pollination filtering as well as the exploration of berberine compound per tissue is supported by the data, although runs short on its discussion. The visitation objective is not fully supported as it is not well described if the reported insects are all the visitors or not, neither if there was time compartamentalization of their visits. The maternal/paternal success objective can be argued as it is a self-pollinating plant with a large pollen to ovule ratio and requires very little pollen for fertilization. The lack of identity of the plant offspring raises a question about the effectivity or necessity of pollinators when offspring genotyping is missing and not much known about zygotic barriers or pollen compatibility.

Strengths

By adopting a 4 species array of plants and insects, it brings strength to a more general function of snapping stamens.

The manipulation experiments to immobilize stamens are a good approach to test the snapping function, with creative methodological variations and additional experiments to discard possible confounding effects.

Weaknesses

The selective in-depth testing of a pair of species makes it harder to generalize the pollen success at the multispecies level.

Although observed in the field (Table S2) provides no further inside observation of the pollinator interactions in nature, their time of activity, or any other insect visitation.

Lack of biological description of the receptivity, fertility, perception, day-cycle, and snapping capacity of the flowers over the 5 days that are assumed to be opened for all species.

No specification is given about some methods, such as the reattachment of manipulated flowers and their age.

Discussion is limited to plant aspects and overlooks insect aspects, such as basic avoidance behaviour (indirect evidence of berberine), possible optimal foraging theory, and learning behaviour (normally assumed in bees). The announced "scarcity of pollen" and self-pollination are lightly discussed and not necessarily based on robust premises. This last is a highly complex theme usually tackled with self-incompatible species due to the difficulty to assign which factor contributes more to forming offspring.

Arguable discussion about dispersal, measured just within the plant. No context on the foraging distances of A. cerana and A. walthoni, or the real impact of pollen compatibility and cross-pollination.

The study can provide a utilitarian reference for other similar questions after certain realization of how general or specific are some conclusions based on the methods and experimental design.

The title states mobile stamens, the use of snapping stamens will be a more accurate term as stated in the manuscript.

Intro

The number of pollen grains is not necessarily few when compared to the number of ovules.

How similar were the insect visitors at each location? And how was it established previously which were visitors and which real pollinators?

Methods

About the visitation and handling time, how are these criteria defined to consider a real visitation and the start and end of handling time?

Why was not tested the snapping in all Berberis species?

Was the evaluation of pollinator interaction in enclosed conditions observed and recorded visually or any camera aid was used?

The self-pollination experiments, have been tried with different genotypes of B. julianae, or how this was assessed? Since the 20m criteria in field condition are quite arbitrary. By not knowing the seed dispersal range or genotype diversity is hard to say if 20m is a reasonable distance.

How was the bee tongue manipulated to make snap the stamens in the experiments with SI/SM?

What is a fine day? Please specify the range of conditions.

How were the removed flowers in the field for SI treatment reattached?

How long flowers of Berberis are open in GH and field? Do reaction time for snapping vary on time?

How was the nectar estimated, by weight or volume, since a 10ul tip does not have so much graduation?

Why were selected the time windows for observation between 9-12 and 12-16, and so on? Is this related to the insect time window of activity?

How to distinguish with certainty selfing from cross-pollination?

In the field experiments with bees, how were other pollinators excluded? and how was recorded the interaction?

Citation missing at the sentence of leptokurtin distribution.

Why the pollen dispersal was evaluated only within the plant and in neighbouring plants?

Was a berberine standard used to confirm the HPLC readings?

Results

The use of "many fewer" can be shortened to just fewer.

What is the scale of vicinity on the flowers observed (f)?

I will suggest taking a look at papers on floral architecture and bee foraging behaviour. And also A. cerana mean flying distance for foraging.

Were other plants evaluated for stained pollen beyond the geitonogamy range?

The berberine response will require a proper electrophysiology test or choice assay or EAG.

Discussion

The proportion of pollen vs ovules, would not be considered scarce.

What about the tasting capacity and learning behaviour of bees with respect to berberine? It can be hypothesized that they will learn to avoid it.

It is mentioned in p.8 that selfed and cross-pollinated B. julianae don't differ, this argues against the pollen scarcity/limitation since very little pollen is needed to fertilize. Are all berberis so highly compatible or does genotype play a role in reducing self-compatibility?

Hard to say an increase in male plant reproductive success by the short distances evaluated.

Tables and figures

How are differences in visitation in Figure 3 A' explained?

Table S1 could profit with a PCA or PCOA.

Table S3 legend, to which species refers? seems a bad modification of the previous table legend.

---

## [Author Response]

Essential revisions:1) The different experiments need to be described in more detail both with regard to the design, procedure, measurements, and analyses. Provide more biological description of the receptivity, fertility, perception, day-cycle, and snapping capacity of the flowers over the 5 days that they are assumed to be opened for all species, including potential auto-deposition by spontaneous snapping.

We have done that and also have clarified which species was observed for how long and where. The experiments are now described more fully and more clearly.

2) Of the four hypotheses the first two have been tested with good experimental designs and statistical power (pollen export from mobile vs. immobile stamens). However, this was not the case for the other two (pollinator filtering, presence or effect of berberine in pollen on pollinators). This should be acknowledged and discussed throughout the manuscript.

As we had written (page 9, lines 355-357, submitted version), “We also did not find a difference between flies and bees in their reaction to the snapping of stamens: Both visitor types left flowers more quickly after being hit by stamens, and stamen snapping therefore did not result in visitor filtering.” Our results thus do not support the hypothesis that mobile stamens contribute to pollinator filtering. We are now stating the three hypotheses tested more clearly in the Introduction and Discussion. Our Table S6 shows the berberine presence in *Berberis* pollen, but not nectar (cf. reply 43), however, any insects’ possible reaction to berberine is not one of the three hypotheses (as now stated more clearly), along with how our data address, support or reject them.

3) Check that the entomology is correct, including species identity and natural history. Also, be careful regarding the interpretation of pollen removal as indicative of berberine content in pollen.

We are very grateful to the reviewer for his comment on the bee shown in Figure 1B. We asked Dr. Michael Orr, an Anthophora expert, to check all our bee photos, and he identified the bee shown in Figure 1B as very likely Habropoda sichuanensis Wu, 1986. This identification was then confirmed (based on pinned specimens) by the first author and another Chinese colleague. Importantly, however, *Anthophora waltoni* is also among the pollinators, based on other pinned specimens and photos. We have corrected this throughout the text. We also consulted Dr. Antti Haarto, Zoological Museum, Section of Biodiversity and Environmental Science, Department of Biology, University of Turku, Finland, an expert on the syrphid genus *Sphaerophoria*, and he determined that the fly for which we had used that name is not a *Sphaerophoria* instead a species of *Meliscaeva*. Our other fly, *Rhingia campestris*, appears correctly identified.

Regarding the berberine in the pollen, please see Table S6 in our manuscript (cf. reply 43).

4) The Discussion should be improved by better synthesizing the different experiments into a general narrative and referring more extensively to general theory and other work on pollination aspects considered in this paper. Also, point out limitations and what would have to be done next to get a more comprehensive and mechanistic understanding of the function of mobile stamens (including pollinator behavior and niche separation as well as pollination results such as increased outcrossing rate or distance).

We now start the Discussion with a short summary about our main findings, and we now end it with a paragraph placing our findings in the context of other work on mobile stamens.

Reviewer #1 (Recommendations for the authors):Besides the main results whether mobile stamens could "filter" visits according to pollinator species identity could not be confirmed and whether berberine affects pollinator behavior could only weakly be inferred from observation of pollen removal from bee tongues. It also remains to be shown whether mobile stamens additionally have a function in ensuring auto-deposition of pollen on stigmas for selfing and whether they increase seed production and offspring fitness of the plant.While the experiments carried out are fine as far as they go, they are somewhat haphazardly planned using different species and places to test different partial questions relevant to the whole story. There is also some intrinsic confounding between different possible functions of mobile stamens, two mentioned by the authors (shorter visit time and more precise placement of pollen on insect body) and additionally the one of auto-deposition of pollen for selfing.The experiments should be described in some more detail and using more precise wording. For example, there are places (e.g. line 257 and lines 295-299) where it is not clear if mobile vs. immobile stamen are compared on the side of the plant from which pollen is exported or on the side of the plant to which pollen is deposited (for example the latter is said in the legend to Figure 5, but from the text, the opposite seems to be the case). It is also not clear if self-pollen reception was tested in the absence or the presence of insects. Would stamens spontaneously snap in older flowers to ensure selfing in case they had not been visited?

Yes, selfing in Berberis is an interesting topic. As we had stated (page 4, lines 166-168, in the original version; now line 180)

“To examine intra-flower selfing induced by the stamen movement, we counted pollen grains deposited per stigma under a light microscope in 16 bagged flowers in which we had triggered one stamen by a needle.”

As shown in Figure S4, spontaneous autogamy does occur in this species, but its fruit/seed sets were both much lower (around 50%) compared to open or outcrossed pollination. We also state clearly (line 253)

“Self-pollen receipt by stigmas of B. julianae after a single stamen movement (14 ± 3 grains, N = 16) was only 6% of the pollen receipt resulting from a single visit by the most common visitors, Apis cerana, which deposited between 230 and 260 grains (section (d)) and roughly 1% of the pollen grains of a single anther with its two pollen sacs (ca. 1220 grains), indicating that intra-flower self-pollination mediated by the stamen movements plays a minor role in total pollen receipt.”

The Discussion section is rather short, reflecting the relatively simple but interesting and novel results. Nevertheless, it seems there must be other literature about the effects of mobile stamens or at least the mechanism could be compared with other mechanisms assumed to reduce ineffective insect pollination in flowering plants or potentially enhance auto-deposition of pollen for selfing (relevant also because there was no difference in pollination success between hand-pollinating selfing and outcrossing treatments). Furthermore, the Discussion could point out where to go from the present "natural history" to more comprehensive studies in the model system. Finally, for this type of study, it would be useful to briefly discuss limitations.

As explained above (reply 4), we now end it with a paragraph placing our findings in the context of other work on mobile stamens, some of it by professor Shuang-Quan Huang, for example, on *Parnassia* (Armbruster et al., 2014).

Generally, the manuscript is well written and the analyses of the simple experiments are fine as far as it can be judged based on the brief descriptions.Reviewer #2 (Recommendations for the authors):This is an interesting report on the functional consequences and adaptive significance of a phenomenon scattered throughout the angiosperm clade: "explosive" snapping movement of stamens in response to being touched. Using four species of Berberidaceae, this study provides the first experimental evidence that rapid stamen movement and active pollen placement increase pollen dispersal to stigmas of other flowers. There is also evidence that the stamen-movement system results in pollen placement being more accurate with respect to sites of stigma contact and pollen being dispensed to a larger number of pollinators. These are all valuable insights into how natural selection has favored the evolution of complex floral behavior.The writing is reasonably clear for the most part, although the sentence structure could be improved in many places. Analyses appear to be appropriate (although more information explaining and justifying the statistical approaches would be desirable). The results and conclusions appear to be valid.Specific comments1. Graphic-abstract caption and in main abstract: "Stamens only snap forward if their filament basis is touched by an insect tongue" implies that artificial stimulation (e.g. with a toothpick) doesn't trigger stamen movement. I don't think you mean this, and if not, rephrase.

Rephrased by dropping “only”.

2. Introduction line 5. "intended" must be in quotes or, better, not used at all.

Replaced by “produced”.

3. Introduction, 1st paragraph. "Flowers are therefore under selection to 'pay' visitors as much as possible by nectar" is potentially misleading. Heinrich and Raven (Science 1972) argued that nectar production/standing crop would not be maximized, but rather optimized to promote pollinator movement.

That’s a misreading of our statement, which is here also quoted out of context. We meant that flowers should ‘pay’ pollinators by nectar instead of pollen (where possible), not that they should produce a lot of nectar.

4. Introduction, 2nd paragraph. I think you mean "base", not "basis".

Corrected, thank you.

5. Materials and methods a, paragraph 2. The preferred word is "pinned" not "needled."

Corrected, thank you.

6. Materials and methods c, 1st sentence. Don't use both passive voice ("were immersed") and active voice ("this blocked") in the same sentence. Change to, e.g., "all stamen movement was blocked…"

Corrected, thank you.

7. Results a, last line. "selfed" is ambiguous. I think you mean "manually self-pollinated".

Corrected, thank you.

8. Results f (and Statistical Methods section). I was unclear as to what assumptions and tests were used to assess the significance of differences in proportions of flowers receiving dyed pollen, e.g. "Of 400 nearby flowers in B. jamesiana, 75 received pollen from SM flowers and 40 from SI flowers, again indicating that mobile stamens donated pollen to more flowers (75/400 vs 40/400, G = 27.81, p < 0.001)." G-test doesn't explain the model or the assumptions, just implies log-likelihood. Is this assuming Normal or Poisson distributions or some nonparametric goodness of fit? The first doesn't seem appropriate and the latter yields a P-value slightly > 0.001 by my quick calculation. This needs a better explanation in the Method section. Because differential dispersal of pollen by motile vs. immobilized stamens is, arguably, the single most important result, its statistical validity should be made crystal clear.

We discussed our statistical approach with a colleague who is teaching biostatistics. He suggested that a G–test of independence (McDonald 2014) would be appropriate, and we now performed a G–test of independence in all comparisons of the effects of stamen movement on pollen flow. The results did not change from what we reported in the earlier version, but we now report the significance based on the G-test.

9. The Discussion would benefit from editing and tighter organization. The language could be improved, removing awkward constructions. The discussion should perhaps start with a brief, well-organized summary of the major results (possibly enumerated), followed by comparisons with other species and studies.

We now start the Discussion with a paragraph enumerating our main results.

Reviewer #3 (Recommendations for the authors):Pollen, containing male gametes of flowering plants, functions in two roles, reproduction for plants and food resources for many insects, especially bees and hoverflies. From the plant aspect, the high fitness should achieve by dispersal pollen grains by the vector to the receptive stigmas of conspecific flowers; for the floral visitor aspect, the high benefit should collect more pollen grains by visiting fewer flowers. This is the pollen dilemma in pollination ecology. Many complex floral traits (concealed pollen grains, pollen/stamen mimicry) or floral behavior (floral part movement, pollen packaging/dispensing) could be driven by solving the "pollen dilemma". In this study, Li et al., presented a well-written paper with well-designed experiments to test the behavior and function of stamen movement in Berberis. The clever experiment design by using alcohol to immobilize stamens made the control and manipulating experiments possible. They found after immobilized stamens, bees visited flowers longer, but removed and deposited fewer pollen grains lowering plant fitness. Therefore, stamen movement drove insects move among flowers, and removed and deposited more pollen grains to conspecific flowers. In my opinion, this is an excellent example to illustrate how "floral trait/behavior" evolves to solve the "pollen dilemma".The authors concluded that "These results demonstrate another mechanism by which plants simultaneously meter out their pollen and reduce pollen theft." I partly agree with this. The evidence to support pollen meter out is very strong, but the evidence to support "reduce pollen theft" is not strong enough.The authors asked four questions, "(1) Are Berberis flowers with mobile stamens visited by the same types of insects and at the same rates as flowers with immobilized stamens? (2) Do flowers with mobile stamens have higher maternal and/or paternal success than those with immobilized stamens? (3) Do mobile stamens contribute to pollinator filtering by eliciting different behaviors in different insect taxa? And (4) is berberine, an alkaloid with antifeedant activity against herbivores that are found in Berberis leaves, also present in Berberis pollen or nectar?", in my opinion, questions (1) and (2) were well addressed in this research. Question (3) was not well addressed by the experiments. Question (4) was descriptive, there is a potential association with questions (1) and (2) but not well explained and tested.Another weakness is the discussion selection. The authors did not fully discuss the novelty and application of key findings to floral trait/behavior evolution in a larger sense. The order of points of discussion also needs to be re-organized following questions and main results. For example, only one sentence talked about floral visitor filtering related to question (3).Using alcohol to immobilize stamens is very important, should expand more to explain how alcohol immobilizes stamen movement in Introduction and Discussion.The time of stamen movement should compare with the flower handling time. Is the time stamen movement always shorter than the time of flower handling by insects?Another limitation of this study is that we still don't know if such floral behavior drive insects to move among individual plant. If such behavior only drives insects to move among flowers within the same plant, it cannot increase the diversity of paternal donors. Since a plant of Berberis can have many many flowers, it is not easy to test plant fitness at an individual plant level. Therefore, it is worth testing the function of stamen movement at individual plant level in the future using a model species producing countable numbers of flowers.

We are very grateful for these many good suggestions! Regarding the time of stamen movement, it is 0.44 ± 0.02 s in *Berberis julianae*, while the flower handling time by *Apis cerana*, the most common pollinator, is 4.919 ± 0.301 s and 5.017 ± 0.661 s by Habropoda sichuanensis (as reported in the manuscript). So, in this species, the stamen movement is much faster than the time that these bee species spent inside a flower. Regarding the genetic consequences of stamen mobility on male fitness, our pollen staining experiments provide empirical evidence for the greater donor distances achieved in flowers with functioning stamens compared to flowers with immobilized stamens. We don’t have genetic data and have therefore removed the word ‘fitness’ from ths manuscript and replaced it with ‘pollen flow distances’ and similar phrases (cf. reply 27)

1. I found Figure 1B is very confusing. I checked Anthophora waltoni, see the following two references, I feel the bee in Figure 1B is so different with A. waltoni in the references. I suggest you should let your entomologist double-check the identification.Mohamed Shebl, Li Qiang, Victor H. Gonzalez "Nesting Behavior, Seasonality, and Host Plants of Anthophora waltoni Cockerell (Hymenoptera: Apidae: Anthophorini) in Yunnan, China," Journal of the Kansas Entomological Society, 87(4), 345-349Zhenghua Xie, Mohamed A. Shebl, Dongdong Pan et al., " Synergistically positive effects of brick walls and farmlands on Anthophora waltoni populations ", Agricultural and Forest Entomology 22(4), pg. 328, (2020); doi:10.1111/afe.12384

Please see reply 2 above. You were correct that the bee on Figure 1B was not *Anthophora waltoni*.

2. This statement on Meliosma tenuis is not accurate, in the case of Meliosma tenuis, very few workers of bumblebees visit flowers, one potential reason is that because workers could not collect the explosively released pollen.

We corrected our statement (line 431) to ‘nectar-seeking bumblebee drones,’ based on Wong Sato and Kato’s conclusion (p. 537):

“The dominant flower visitors were nectar-seeking drones of the bumblebee species *Bombus ardens* (Apidae). The drone’s behavior, pollen attachment on their bodies, and fruit set of visit-restricted flowers suggest that they are the only agent triggering the explosive pollen release mechanism, and are the main pollinator of *M. tenuis*.”

3. "Flowers are therefore under selection to 'pay' visitors as much as possible by nectar". Don't drop into the trap that "nectar is for the goodness of floral visitor", nectar production is selected for the plant's own fitness, but such fitness is realized by floral visitation.

We did not fall into any trap about nectar production (this assumption by the reviewer is almost condescending). Please see reply 9.

4. "Replenishable nectar". However, not all plant species replenish nectar, see the following reference. Luo, E.Y., Ogilvie, J.E. and Thomson, J.D. (2014). Stimulation of flower nectar replenishment by removal: A survey of eleven animal-pollinated plant species. J. Pollin. Ecol., 12, 52-62.

We have now added “usually” before “replenishable.” The point here is that pollen is non-replenishable – ever. We agree with the reviewer’s general point, however, and one of us has even published on rewardless flowers: Renner, S. S. 2006. Rewardless flowers in the angiosperms and the role of insect cognition in their evolution. In N. M. Waser and J. Ollerton (eds.), Plant-Pollinator Interactions: From Specialization to Generalization, pp. 123-144. Univ. of Chicago Press, Chicago.

Reviewer #4 (Recommendations for the authors):The study brings attention to a long-standing and assumed function of snapping stamens in Berberis spp. It contains four objectives to which the methodological approach is standard and most are broadly supported in the literature, even though the suitability of some designs could be argued as suboptimal proof of concept for generalization of certain pollination conclusions. This works aims to prove that visitation doesn't differ when stamens are able or not of snapping, but that the snapping mechanism improves the pollen uptake and transfer to pollinators and thus improves fruit formation, while the signature alkaloid compound of Berberis julianae is not a detriment to the interaction.The results support well the null hypothesis of pollination filtering as well as the exploration of berberine compound per tissue is supported by the data, although runs short on its discussion. The visitation objective is not fully supported as it is not well described if the reported insects are all the visitors or not, neither if there was time compartamentalization of their visits. The maternal/paternal success objective can be argued as it is a self-pollinating plant with a large pollen to ovule ratio and requires very little pollen for fertilization. The lack of identity of the plant offspring raises a question about the effectivity or necessity of pollinators when offspring genotyping is missing and not much known about zygotic barriers or pollen compatibility.StrengthsBy adopting a 4 species array of plants and insects, it brings strength to a more general function of snapping stamens.The manipulation experiments to immobilize stamens are a good approach to test the snapping function, with creative methodological variations and additional experiments to discard possible confounding effects.WeaknessesThe selective in-depth testing of a pair of species makes it harder to generalize the pollen success at the multispecies level.Although observed in the field (Table S2) provides no further inside observation of the pollinator interactions in nature, their time of activity, or any other insect visitation.Lack of biological description of the receptivity, fertility, perception, day-cycle, and snapping capacity of the flowers over the 5 days that are assumed to be opened for all species.No specification is given about some methods, such as the reattachment of manipulated flowers and their age.Discussion is limited to plant aspects and overlooks insect aspects, such as basic avoidance behaviour (indirect evidence of berberine), possible optimal foraging theory, and learning behaviour (normally assumed in bees). The announced "scarcity of pollen" and self-pollination are lightly discussed and not necessarily based on robust premises. This last is a highly complex theme usually tackled with self-incompatible species due to the difficulty to assign which factor contributes more to forming offspring.Arguable discussion about dispersal, measured just within the plant. No context on the foraging distances of A. cerana and A. walthoni, or the real impact of pollen compatibility and cross-pollination.The study can provide a utilitarian reference for other similar questions after certain realization of how general or specific are some conclusions based on the methods and experimental design.The title states mobile stamens, the use of snapping stamens will be a more accurate term as stated in the manuscript.

We changed the title to “Touch-sensitive stamens enhance pollen dispersal by scaring away visitors”

IntroThe number of pollen grains is not necessarily few when compared to the number of ovules.

Good point. We now provide the ratio of pollen grains to ovules in the studied species, viz. 2379 ± 49 in *Berberis julianae* and 3405 ± 78 in *Berberis jamesiana*.

How similar were the insect visitors at each location? And how was it established previously which were visitors and which real pollinators?

As stated, we studied

*1) Berberis julianae* C.K.Schneider in a field located at 29°52′26″N, 105°30′32″E, 427 m above sea level, in southeast of Anyue County, Sichuan Province, China.

*2) B. jamesiana* Forrest and W.W.Smith at Shangri-La Alpine Botanical Garden (N 27°54′05″, E 99°38′17″, 3300-3350 m above sea level), in Yunnan Province, Southwestern China.

*3) B. forrestii* Ahrendt at the Shangri-La field station,

*4) Mahonia bealei* (Fortune) Carrière in the Wuhan Botanical Garden (N 30°33′2″, E 114°25′48″, 23 m above sea level) in Hubei Province, China.

Given that each species was studied at a different location, we did not focus on comparing visitor spectra. There was no need to establish a priori which insects were visitors and which were pollen vectors because we have empirical observations on this point (especially given our experiments with stained pollen that was tracked among plants).

MethodsAbout the visitation and handling time, how are these criteria defined to consider a real visitation and the start and end of handling time?Why was not tested the snapping in all Berberis species?

As stated on page 5, line 213-219, and shown in Table S1, we tested the stamen bending time in all three *Berberis* species as well as the *Mahonia* species.

Was the evaluation of pollinator interaction in enclosed conditions observed and recorded visually or any camera aid was used?

We produced videos of the interactions, three of which accompany our paper.

The self-pollination experiments, have been tried with different genotypes of B. julianae, or how this was assessed? Since the 20m criteria in field condition are quite arbitrary. By not knowing the seed dispersal range or genotype diversity is hard to say if 20m is a reasonable distance.

Yes, our experiments might involve genetic siblings. Assessing genotypes and sib mating in *Berberis* was beyond the scope of our study. Please compare reply 16.

How was the bee tongue manipulated to make snap the stamens in the experiments with SI/SM?

To quantify pollen loads placed on insect bodies and stigmas of next-visited conspecific flowers, we held captured and anaesthetized bees between forceps and made their tongues contact the filament bases in SI or SM flowers. This was described in the Supplementary Methods, but we now also described it in the main text’s M and M section.

What is a fine day? Please specify the range of conditions.

A sunny day (now stated, in response to reviewer #3).

How were the removed flowers in the field for SI treatment reattached?

As shown in Figure 1H, we inserted the pedicels of SI flowers into an inflorescence. We now state this clearly in the M and M section of the main text

How long flowers of Berberis are open in GH and field? Do reaction time for snapping vary on time?

Each flower lasts for about 4-5 days in Berberis julianae. We did not study changes in the stamen reaction ( = bending) time over several days, but do report Inward movement time (s), Interval time (s), and Outward movement time (s) in Table S1.

How was the nectar estimated, by weight or volume, since a 10ul tip does not have so much graduation?

The nectar volume per flower of *B. julianae* was estimated by using clean 10 µL microliter syringes (Agilent Technologies Inc, USA) to extract nectar drops from bagged flowers. This was and is described in the Suppl. Mats.

Why were selected the time windows for observation between 9-12 and 12-16, and so on? Is this related to the insect time window of activity?

Yes, the observation times reflect the insects’ main activity times.

How to distinguish with certainty selfing from cross-pollination?

Selfing refers to receipt of a flower’s own pollen grains. Cross-pollination refers to receipt of pollen grains from a different *Berberis* shrub. As explained above, data on genetic structure (and possible sib mating) were beyond the scope of this study.

In the field experiments with bees, how were other pollinators excluded? and how was recorded the interaction?

In the field experiments with bees, we did not exclude other pollinators, since there were too few non-bee visits to visit the treated flowers (which in one experiment were harvested after single bee visits; as described in M and M).

Citation missing at the sentence of leptokurtin distribution.

We now added a review to support our statement “Previous studies have indicated that pollen flow mediated by generalist insects usually occurs within meters with a highly leptokurtic distribution.” Williams, 2001, which concluded “Studies of pollen and gene flow within patches/crops [..-], whether wind- or insect-pollinated, generally describe highly leptokurtic distributions of pollen from the source plants, with levels decreasing rapidly within a few metres of the source to a low level at which they remain for a much longer distance.

Why the pollen dispersal was evaluated only within the plant and in neighbouring plants?

Because detecting the stained pollen grains in the vicinity of the donor plant was difficult. We checked hundreds of stigmas at the distances mentioned in the manuscript and Tables S3 and S4.

Was a berberine standard used to confirm the HPLC readings?

Yes, a berberine standard was used to confirm the HPLC readings.

ResultsThe use of "many fewer" can be shortened to just fewer.

Done.

What is the scale of vicinity on the flowers observed (f)?

About 25 to 100 cm from the source as indicated in tables S3 and S4. We have now added this information in the sentence where the word ‘vicinity’ first appears.

I will suggest taking a look at papers on floral architecture and bee foraging behaviour. And also A. cerana mean flying distance for foraging.

Thank you for this suggestion.

Were other plants evaluated for stained pollen beyond the geitonogamy range?

We checked hundreds of stigmas (specifically up to 994 and 733) on different shrubs at the distances mentioned in the manuscript and Tables S3 and S4.

The berberine response will require a proper electrophysiology test or choice assay or EAG.

Why? We observed bees cleaning Berberis pollen from their bodies, and this is all that we report, along with the berberine concentrations in leaves, flowers, pollen, and nectar (Table S6).

DiscussionThe proportion of pollen vs ovules, would not be considered scarce.

We revised as **“**Especially in flowers with open-access pollen grains”.

What about the tasting capacity and learning behaviour of bees with respect to berberine? It can be hypothesized that they will learn to avoid it.

We agree, but have no data on these points.

It is mentioned in p.8 that selfed and cross-pollinated B. julianae don't differ, this argues against the pollen scarcity/limitation since very little pollen is needed to fertilize. Are all berberis so highly compatible or does genotype play a role in reducing self-compatibility?

These interesting questions are outside the scope of our study.

Hard to say an increase in male plant reproductive success by the short distances evaluated.

We beg to differ. In flowers with immobilized stamens, the commonest bee species stayed up to 3.6x longer, yet removed 1.3x fewer pollen grains and deposited 2.1x fewer grains on stigmas per visit. Mobile stamens exported their pollen to significantly more flowers.

Tables and figuresHow are differences in visitation in Figure 3 A' explained?

Figure 3 A' shows the visitation rates of *Apis cerana* to SM and SI flowers in the field. As described in M and M, SI flowers were cut off and their pedicels inserted into little vials filled with alcohol, and after a certain time, this probably reduced their nectar production.

Table S1 could profit with a PCA or PCOA.

Table S1 floral traits (mean ± SE) and duration of stamen movements in *Berberis jamesiana*, *B. julianae*, *B. forrestii*, and *Mahonia bealei*. Different superscript letters indicate significant differences; N = number of sampled flowers; － not measured. We do not understand why a principal component analysis is needed here.

Table S3 legend, to which species refers? seems a bad modification of the previous table legend.

We have dropped Table S3.